# Caldesmon controls stress fiber force-balance through dynamic cross-linking of myosin II and actin-tropomyosin filaments

Shrikant B. Kokate [1], Katarzyna Ciuba[1,5], Vivien D. Tran[2], Reena Kumari[1], Sari Tojkander[3], Ulrike Engel[4], Konstantin Kogan [1], Sanjay Kumar [2] & Pekka Lappalainen [1] ✉

Contractile actomyosin bundles are key force-producing and mechanosensing elements in muscle and non-muscle tissues. Whereas the organization of muscle myofibrils and mechanism regulating their contractility are relatively well-established, the principles by which myosin-II activity and force-balance are regulated in non-muscle cells have remained elusive. We show that Caldesmon, an important component of smooth muscle and non-muscle cell actomyosin bundles, is an elongated protein that functions as a dynamic cross-linker between myosin-II and tropomyosin-actin filaments. Depletion of Caldesmon results in aberrant lateral movement of myosin-II filaments along actin bundles, leading to irregular myosin distribution within stress fibers. This manifests as defects in stress fiber network organization and contractility, and accompanied problems in cell morphogenesis, migration, invasion, and mechanosensing. These results identify Caldesmon as critical factor that ensures regular myosin-II spacing within non-muscle cell actomyosin bundles, and reveal how stress fiber networks are controlled through dynamic cross-linking of tropomyosin-actin and myosin filaments.

Contractile actomyosin bundles, composed of thin actin filaments and thick myosin II filaments, are key force producers and mechanosensors in diverse animal cell types and tissues. They include myofibrils of striated muscles, contractile filaments of smooth muscles, and stress fibers of non-muscle cells[1,2]. Myofibrils are comprised of regular, repetitive patterns of the force-generating and load-bearing devices, called sarcomeres. The force for muscle contraction is generated by ATP-dependent sliding of bipolar myosin II filaments along a bipolar array of actin filaments within the sarcomeres. Muscle contraction is activated by $Ca^{2+}$ released from the sarcoplasmic reticulum through the troponin–tropomyosin (Tpm) complex. $Ca^{2+}$-binding to troponin induces a conformational change, which shifts the position of Tpm

along the major groove of the actin filament to allow actin–myosin interactions and muscle contraction[3]. Two structures within the sarcomeres, the Z-disc and the M-band, anchor actin and myosin II filaments to an elastic filament system that is composed of giant protein titin. The Z-disc harbors α-actinin cross-linked antiparallel actin filaments from the adjacent sarcomeres and several associated proteins[4,5]. M-band at the center of the sarcomere is composed of a hexagonal array of proteins including myomesins that position the myosin II filaments to the middle of the sarcomere, and are hence critical in controlling the force imbalances during muscle contraction[6].

Whereas the architecture of striated muscle myofibrils is relatively well understood, much less is known about the contractile actomyosin

[1]HiLIFE Institute of Biotechnology, University of Helsinki, P.O. Box 56, 00014 Helsinki, Finland. [2]Department of Bioengineering, University of California, Berkeley, CA 94720, USA. [3]Faculty of Medicine and Health Technology, Tampere University, Kauppi Campus, Arvo Building, E318, Arvo Ylpön katu 34, 33520 Tampere, Finland. [4]Nikon Imaging Center at Heidelberg University and Centre for Organismal Studies (COS), Heidelberg University, Im Neuenheimer Feld 267, Heidelberg 69120, Germany. [5]Present address: Nencki Institute of Experimental Biology PAS, 3 Pasteur Street, 02-093 Warszawa, Poland. ✉e-mail: pekka.lappalainen@helsinki.fi

bundles of smooth muscle and non-muscle cells. Stress fibers are key mechanosensing elements in non-muscle cells, and they share many structural components with muscle sarcomeres[7–11]. However, smooth muscle and non-muscle cells lack many key components of striated muscle myofibrils, such as proteins of the M-band. Therefore, how the organization of actomyosin bundles within contractile units is maintained, and how the force imbalances during contraction are controlled in smooth muscles and non-muscle cells have remained elusive[1,2]. Moreover, stress fibers and smooth muscle actomyosin bundles do not express the troponin complex, and thus the mechanisms controlling their contractility are incompletely understood.

Although stress fibers and smooth muscle actomyosin bundles lack certain key components of striated muscle myofibrils, they contain other proteins that may functionally replace central striated muscle components. These include actin/myosin-binding protein Caldesmon, which has smooth muscle (H-Caldesmon) and non-muscle cell (L-Caldesmon) specific splice variants. Apart from the ~250 residue regions in the center of H-Caldesmon, the two splice variants are identical to each other[12,13]. The N-terminal ~200 residue region of Caldesmon interacts with both myosin II and calmodulin[14], and the C-terminal ~200 residue region binds actin filaments, tropomyosin, and calmodulin[15–18]. The C-terminal region of Caldesmon inhibits the actin-activated ATPase of myosin II in vitro, and the binding of calmodulin to this region decreases its affinity for F-actin and releases the inhibition[19–21]. Based on biochemical studies, several different functions for Caldesmon were proposed. According to the popular view, Caldesmon regulates actomyosin contractility in a $Ca^{2+}$/calmodulin and phosphorylation-dependent manner and is hence functionally equivalent to the skeletal muscle troponin complex[13]. Caldesmon was additionally proposed to stabilize actin filaments to maintain the structural integrity of actomyosin bundles[22,23], and stabilize the latch-state of myosin II capable of maintaining force for long periods with low energy consumption[24].

Despite extensive biochemical data, the in vivo function of Caldesmon has remained elusive. Knockout mice, lacking both L-Caldesmon and H-Caldesmon, die perinatally[25], whereas specific deletion of the H-Caldesmon resulted in defects in arterial muscle relaxation[26]. In non-muscle cells, Caldesmon over-expression increases the thickness of stress fibers in lung carcinoma A549 cells, and depletion of Caldesmon by RNAi resulted in thinner stress fibers[27]. On the other hand, over-expression of Caldesmon decreased the contractility of SV80 cells[28]. Whether these phenotypes result from defects in actomyosin bundle organization, increased contractility of actomyosin bundles, from the loss of myosin II latch-state, or from some other defects in actomyosin bundles is not known. Altered expression levels of Caldesmon are also linked to the progression of many cancers[29,30] and developmental abnormalities[25,31–33], but the underlying mechanism has remained elusive.

Here, we revealed that Caldesmon is an extended protein that displays a highly dynamic association with stress fibers. Importantly, studies on Caldesmon knockout and rescue cells provide evidence that Caldesmon does not function as a negative regulator of myosin II activity, but that it instead cross-links myosin filaments and actin-tropomyosin filaments to maintain the regular spacing of myosin II filaments within actomyosin bundles. Our findings reveal how force imbalances in actomyosin bundles, and the organization of the mechanosensitive stress fiber networks, are controlled through dynamic cross-linking of actin and myosin filaments.

## Results

### Caldesmon is a dynamic component of contractile stress fibers

Caldesmon associates with stress fibers in non-muscle cells[28,34], but its precise localization within actomyosin bundles has not been reported. We applied 3D super-resolution structured illumination microscopy (3D-SIM) on U2OS cells immunostained with Caldesmon-specific antibody, and on cells expressing mCherry-fusion of Caldesmon. Both endogenous Caldesmon and mCherry-Caldesmon displayed periodic localization patterns along the contractile ventral stress fibers and transverse arcs but were absent from the non-contractile dorsal stress fibers (Fig. 1a; Supplementary Fig. 1a, b). Caldesmon localization appears to be independent of rapid actin dynamics because diminishing actin filament assembly by Latrunculin-A treatment did not displace Caldesmon from stress fibers (Supplementary Fig. 1c).

Several proteins, such as NM-IIA, NM-IIB, myosin-18B, and tropomyosins show relatively stable association with stress fibers, whereas others including palladin, α-actinin, VASP, and calponin-3 display rapid dynamics at stress fibers[7,9,35–37]. Fluorescence recovery after photobleaching (FRAP) experiments on ventral stress fibers of U2OS cells expressing either GFP-Caldesmon or GFP-NM-IIA revealed that Caldesmon is a highly dynamic stress fiber component, with halftime of approximately 1–2 s. As reported earlier[7,38], the recovery of NM-IIA heavy-chain was much slower compared to Caldesmon (Fig. 1b, d, e). Because Caldesmon was proposed to regulate myosin II activity in vitro[21], we also tested the possible effects of myosin II inhibition on Caldesmon dynamics. Inhibiting the ATPase activity of myosin II by para-amino-blebbistatin[39] did not result in detectable effects on Caldesmon dynamics at stress fibers (Fig. 1c and f). Together, these experiments show that Caldesmon localizes to myosin II-containing, contractile stress fibers and is a highly dynamic component of these actomyosin bundles.

### Caldesmon is an extended molecule that cross-links the necks of myosin II filaments to tropomyosin-actin filaments

To reveal the precise localization of Caldesmon, and its different protein domains within stress fibers, we performed 3D-SIM on U2OS cells expressing a Caldesmon construct fused to mCherry at its N-terminus and to GFP at its C-terminus. Interestingly, the N- and C-terminal mCherry and GFP of Caldesmon did not co-localize with each other at stress fibers, but were separated from each other by ~0.3 μm (Fig. 2a, b). This provides evidence that in cells Caldesmon is an extended protein. We next examined the relationship of Caldesmon and myosin II in more detail by staining mCherry-Caldesmon expressing U2OS cells with NM-IIA coiled-coil and NM-IIA regulatory light-chain (RLC)-specific antibodies. 3D-SIM analysis revealed that the N-terminal region of Caldesmon localized to the neck region of myosin II. This is because the mCherry signal displayed two peaks that flanked the NM-IIA coiled-coil signal (Fig. 2c and e), but was located in between the two myosin motor domain signal peaks within individual NM-IIA bundles (Fig. 2d and f).

To uncover the mechanisms by which Caldesmon localizes to stress fibers, we generated truncated versions of Caldesmon, and expressed those as N-terminal mCherry-fusions in U2OS cells. The N-terminal domain of Caldesmon (residues 1–200), which binds myosin II in vitro[13] and associates with the necks of myosin II bundles in cells (Fig. 2c–f) displayed diffuse cytoplasmic localization with only very weak enrichment to stress fibers. On the other hand, the C-terminal region of Caldesmon (residues 380–501) displayed strong localization to stress fibers. Moreover, deletion of the C-terminal region resulted in diffuse cytoplasmic localization of Caldesmon (Supplementary Fig. 2a, b). However, unlike the full-length Caldesmon, the C-terminal region (residues 380–501) localized also to non-contractile dorsal stress fibers (Supplementary Fig. 2c) and displayed relatively uniform/random localization along the contractile ventral stress fibers (Supplementary Fig. 2d, e). The expression of these constructs did not alter the overall cell shape or stress fiber architecture in wild-type U2OS cells.

Because the C-terminal region of Caldesmon binds F-actin[40] and interaction of the full-length smooth muscle Caldesmon to actin filaments was reported to be enhanced by Tpm[19], we examined the effects of different stress fiber-associated Tpms on actin

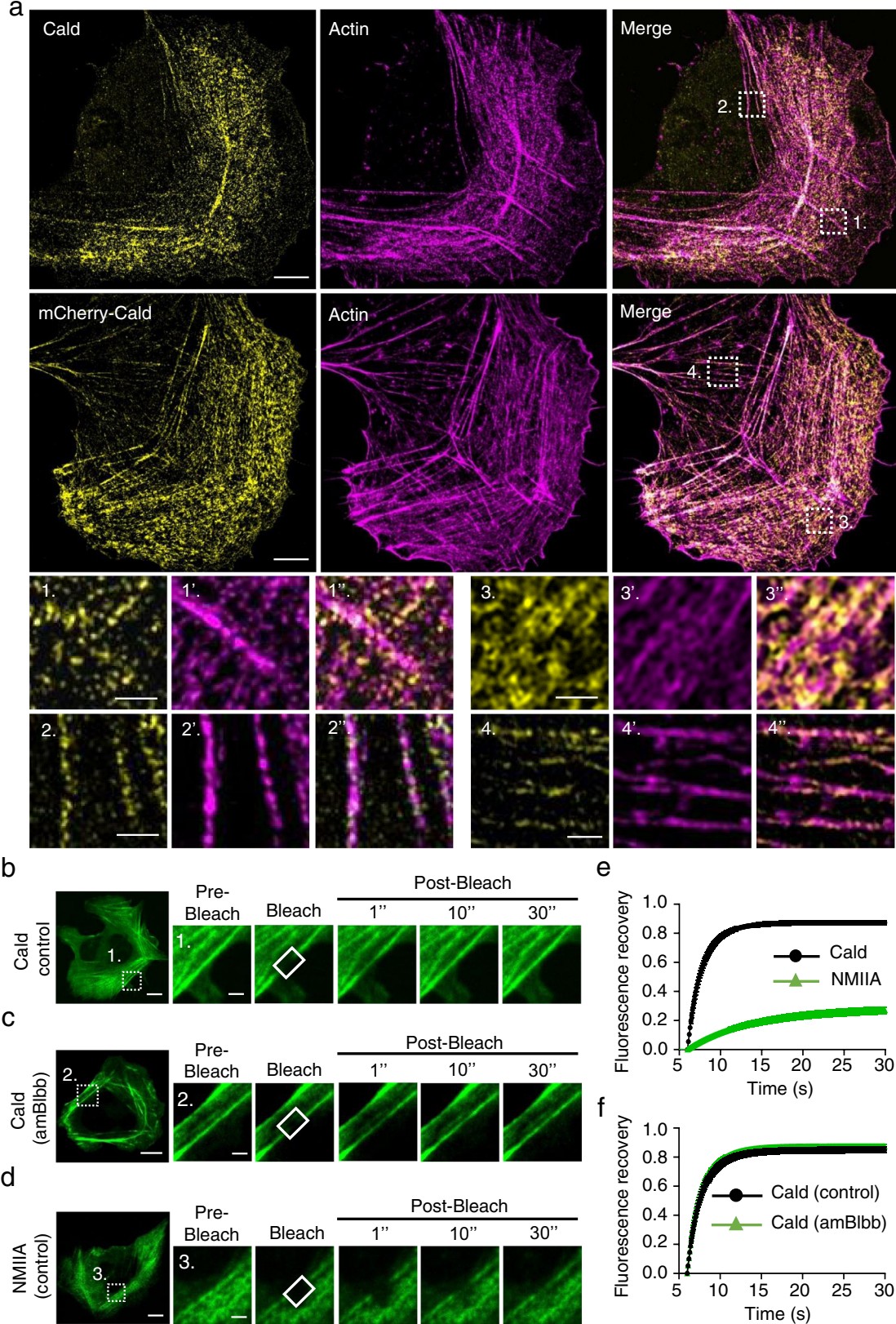

interactions of Caldesmon. Actin filament co-sedimentation assay revealed that both full-length Caldesmon and its C-terminal region (residues 380–501) bind actin filaments, with the affinity of the full-length Caldesmon being slightly higher (Supplementary Fig. 3). Binding of both full-length Caldesmon and its C-terminal region were enhanced by saturating actin filaments with Tpm1.6, Tpm2.1 and Tpm3.2, whereas decoration of actin filaments with Tpm4.2 had no detectable effect on Caldesmon binding (Supplementary Fig. 2f, g). Consistent with the biochemical data, the C-terminal domain of Caldesmon displayed slightly less co-localization with Tpm4.2 in U2OS cells as compared to other stress fiber–associated Tpm iso-forms (Supplementary Fig. 3g).

**Fig. 1 | Caldesmon is a dynamic component of contractile stress fibers.**
**a** Representative 3D-SIM image demonstrates that Caldesmon localizes to contractile ventral stress fibers and transverse arcs, but is absent in non-contractile dorsal stress fibers in U2OS cells. Endogenous Caldesmon was visualized by specific antibody (upper row), and mCherry-Caldesmon construct was applied to confirm the subcellular localization of the protein (second row). Actin filaments we visualized by fluorescent phalloidin. Panels at the bottom are magnifications of the regions indicated with white boxes in the upper panels. From these, 1, 1' and 1" show absence of endogenous Caldesmon at dorsal stress fibers, 2, 2' and 2" localization of endogenous Caldesmon to ventral stress fibers, 3, 3' and 3" localization of mCherry-Caldesmon to transverse arcs, and 4, 4' and 4" localization of mCherry-Caldesmon to cortical stress fibers. The images summarize observations from at least three independent experiments. Scale bars, 5 and 1 μm in whole cell images and magnified images, respectively. **b–d** Representative examples of fluorescence-recovery-after-photobleaching (FRAP) analysis. Dynamics GFP-Caldesmon in ventral stress fibers of an untreated wild-type U2OS cell (panel **b**), a 5 μM *para*-amino-blebbistatin-treated U2OS cell (panel **c**), as well as dynamics of GFP-NM-IIA heavy chain in a ventral stress fiber of a wild-type U2OS cell (panel **d**). **e** Recovery curves demonstrate that Caldesmon is a highly dynamic component of stress fibers as compared to NMII-A. **f** Recovery curves of GFP-Caldesmon in control and *para*-amino-blebbistatin-treated cells provide evidence that NM-IIA inhibition does not affect the dynamics of Caldesmon. Graphs show mean ± SEM from *n* = 21 (Caldesmon control), 18 (Caldesmon blebbistatin), and 23 (NM-IIA control) cells from three experiments. Scale bars, whole cells 15 μm, magnified time-lapse images 5 μm.

Together, these results reveal that Caldesmon is an extended molecule in cells. Caldesmon associates with the neck region of myosin II and with Tpm-decorated actin filaments through its N-terminal domain and C-terminal domain, respectively.

## Caldesmon knockout cells display defects in cell morphology and migration

To uncover the cellular function of Caldesmon, we generated Caldesmon knockout U2OS cell lines using CRISPR/Cas9 approach. Following cell sorting (Supplementary Fig. 4a), Sanger and NGS sequencing were performed for two knockout clones generated by two different guide-RNAs. This confirmed complete Caldesmon inactivation in both knockout clones (Supplementary Fig. 4b–f). Both sequenced clones also displayed depletion of Caldesmon protein as determined by Western blotting and by immunofluorescence microscopy (Fig. 3a and Supplementary Fig. 5a).

Visualizing F-actin by fluorescent phalloidin and focal adhesions by vinculin-specific antibody in cells grown on fibronectin-coated coverslips and on crossbow micropatterns revealed that stress fibers were still present in the knockout cells, but the stress fiber network was less regularly organized as compared to wild-type U2OS cells. The dorsal stress fibers in knockout cells were typically abnormally long, and transverse arcs often displayed a 'tilted' orientation compared to the leading edge of cell (Fig. 3b and Supplementary Fig. 5a). Moreover, Caldesmon knockouts displayed larger cell area as compared to wild-type cells, and were more elongated and irregular-shaped (Fig. 3c, d). These phenotypes could be partially rescued by over-expression of full-length GFP-Caldesmon, demonstrating that they do not result from off-target effects. Caldesmon depletion was also accompanied by defects in cell motility, as measured from the migration of individual cells plated on fibronectin-coated surfaces, and collective migration as measured by wound healing assay (Fig. 3e–h; Supplementary Movies 1–3). Finally, the transwell matrigel assay revealed a significant decrease in invasive migration of Caldesmon knockout cells (Supplementary Fig. 5b, c). These results demonstrate that Caldesmon is not critical for stress fiber assembly or maintenance, but its depletion results in abnormal organization of the stress fiber network, as well as defects in cell morphogenesis, migration, and invasion.

## Caldesmon is not a negative regulator of stress fiber contractility

Based on biochemical data, Caldesmon was considered to function as a negative regulator of myosin II[13]. Hence, it is expected that Caldesmon depletion would result in increased contractility of stress fibers. To examine the role of Caldesmon in stress fiber contractility, we first applied traction-force microscopy (TFM) on wild-type and Caldesmon knockout cells. Surprisingly, these experiments revealed that Caldesmon knockout cells exert 50–75% diminished contractile forces onto the substrate as compared to the wild-type U2OS cells (Fig. 4a, b, and Supplementary Fig. 5d, e). We next examined the contractility of stress fibers by monitoring the retrograde flow of transverse arcs. These actomyosin bundles are assembled at the lamellipodia–lamellum

interface, and they undergo myosin II-dependent retrograde flow towards the cell center[7,41]. During retrograde flow, transverse arcs can fuse with each other to form thicker actomyosin bundles that mature into ventral stress fibers[42]. Live-cell imaging of wild-type and Caldesmon knockout cells, transfected with a plasmid expressing GFP-Lifeact, on a fibronectin-coated surface indicated that the knockout cells display slower retrograde flow of transverse arcs, and defects in their fusion with each other (Supplementary Fig. 5f, g; Supplementary Movies 4 and 5). To study this in a more controlled environment, we utilized fibronectin-coated circular micropatterns. Live imaging of cells plated on circular fibronectin islands revealed that the dynamic behavior of stress fibers was indeed altered in Caldesmon-depleted cells (Supplementary Movies 6 and 7). Although the retrograde flow of transverse arcs was still evident also in the Caldesmon knockout cells, the rate of transverse arc flow was diminished by ~40% (Fig. 4c, d).

Nuclear localization of downstream transducers of the Hippo pathway, YAP/TAZ, are regulated by substrate rigidity and contractile forces of the cell[43,44]. We thus compared nuclear localization of YAP in wild-type and Caldesmon knockout cells plated on 8 and 50 kPa matrices. Please note that in these experiments Caldesmon was visualized by mouse polyclonal antibody, which also displays non-specific staining in the nuclear and perinuclear regions. When plated on 8 kPa matrix, both wild-type and knockout cells exhibited similar, predominantly cytoplasmic YAP localization. However, in cells plated on 50 kPa matrix, YAP displayed more profound nuclear localization in wild-type cells as compared to Caldesmon knockout cells (Fig. 4e–g). Thus, Caldesmon knockout cells have defects in sensing substrate stiffness.

To understand in more detail how depletion of Caldesmon affects the mechanical properties of individual stress fibers, we executed laser nanosurgery experiments on transverse arcs of wild-type and Caldesmon knockout cells (Fig. 5a–f). Following our previous studies[45–48], we visualized stress fibers in wild-type and Caldesmon knockout cells through lentiviral transduction with RFP-Lifeact. To standardize cell shape and size and induce strong transverse arc formation, we utilized single-cell micropatterned glass coverslips in crossbow shapes that mimic polarized cell shapes. We then used femtosecond laser ablation to sever selected transverse arcs and fit the resulting time-dependent retraction to a Kelvin–Voigt (KV) model to obtain a time constant ($\tau$) and plateau retraction distance ($L_o$). In some cases, arc retraction did not fit KV retraction kinetics, perhaps because of the disconnect between the curved morphology of the arcs and the linear structure assumed by the KV model. We therefore subjected all SFs to a second, model-independent analysis in which we measured the distance retracted by one end of a cut fiber 60 s after ablation (Fig. 5b).

We did not observe statistically significant differences across the two cell populations in $\tau$, $L_o$, or retraction at 60 s, implying that Caldesmon does not measurably affect the mechanics of individual transverse arc fibers (Fig. 5a–f). To provide greater standardization of cell shape and stress fiber length while investigating the mechanics of ventral stress fibers, we revisited these studies on rectangular frame micropatterns (Supplementary Fig. 6a–e). Here, a short gap is introduced on one edge of the frame to promote the assembly of a ventral

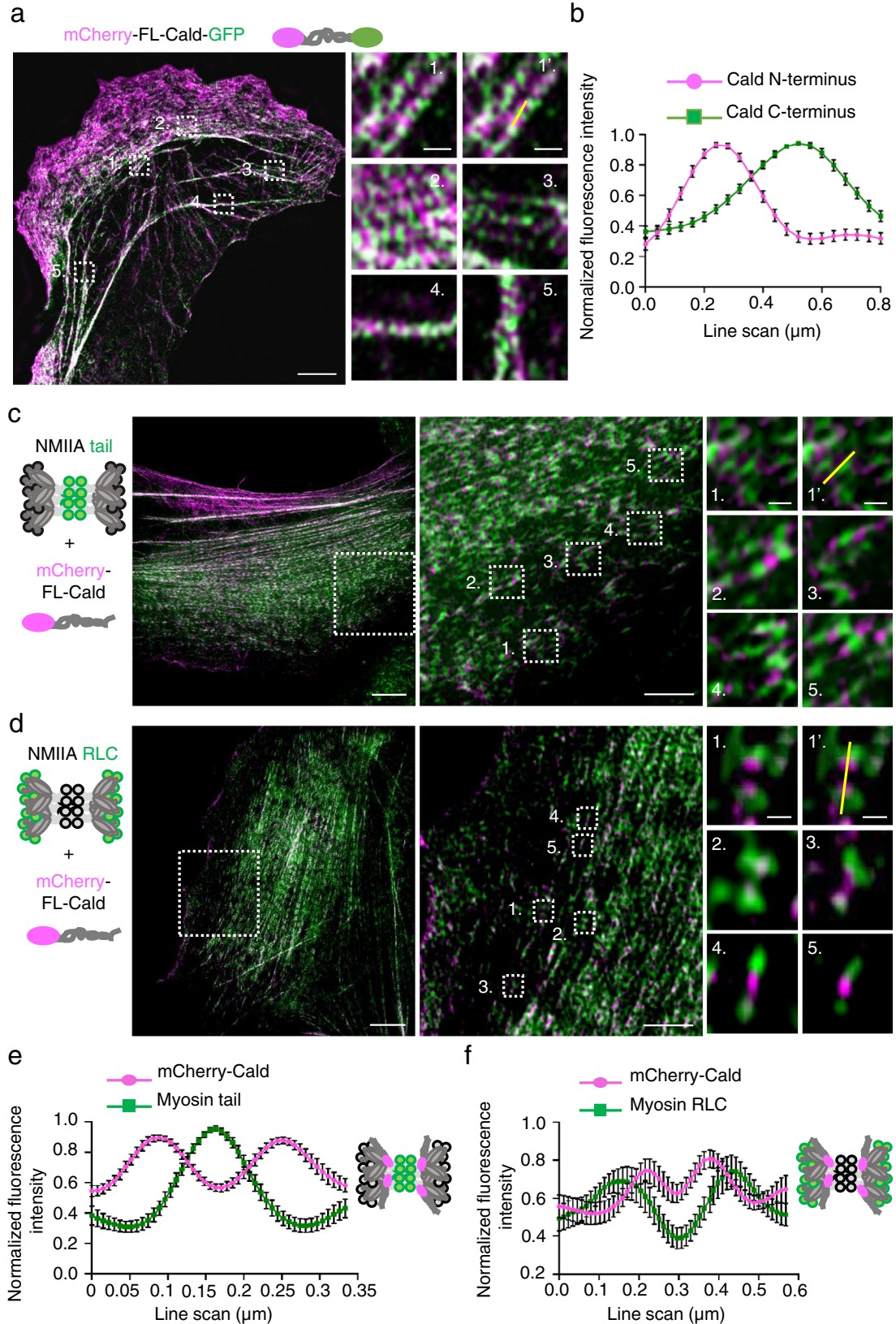

stress fiber of defined length, which could then be targeted for laser ablation. This geometry also did not yield significant Caldesmon-dependent differences in single-stress fiber mechanics. Similar results were obtained through laser ablation experiments repeated on cells cultured on unpatterned, fibronectin-coated glass (Supplementary Fig. 6f–i).

Together, these experiments demonstrate that depletion of Caldesmon does not result in increased stress fiber contractility, as would be expected if it would function as a negative regulator of myosin II as previously proposed. Instead, we show that the mechanics of individual stress fibers are largely unaltered in Caldesmon knockout cells, whereas the organization of the stress fiber network and overall forces

**Fig. 2 | Caldesmon is an extended protein and its N-terminal domain associates with the neck of myosin II filament. a** Representative example of a 3D-SIM image obtained from U2OS cell expressing full-length Caldesmon in which mCherry (magenta) and GFP (green) were fused to the N- and C-termini of the protein, respectively. The panels on the right are magnifications of the regions indicated with white boxes in the whole cell image. The yellow line in panel 1' shows an example of a region used for line scans. Scale bars, whole cell 10 μm, magnified images 2 μm. **b** Line plot analysis of mCherry-Cald-GFP indicates that the mean distance between N- and C-terminal GFP and mCherry of Caldesmon is ~0.3 μm. Line plot shows the mean of normalized fluorescence intensity ± SEM of $n = 30$ filaments from three cells. **c** Representative 3D-SIM image obtained from a U2OS cell expressing mCherry-Caldesmon (magenta) and stained with NM-IIA tail-specific antibody (green). White boxes indicate the magnified regions. The magnified image 1' shows an example of an NM-IIA filament/Caldesmon that was used for line plot analysis. **d** Representative 3D-SIM image of U2OS cell expressing mCherry-Cald (magenta) and stained with an antibody specific to NM-IIA RLC (green). White boxes indicate the magnified regions. The magnified image 1' shows an example of a NM-IIA filament that was used for line plot analysis. Scale bars for panels **c** and **d**, 5, 2.5, and 1 μm for the left, middle, and right panels, respectively. **e** Line plot analysis of the co-localization of Caldesmon N-terminus and NM-IIA tail. Line scans = mean ± SEM of $n = 15$ filaments from three cells. **f** Line plot profile showing co-localization of Caldesmon N-terminus and NM-IIA RLC. Line scans = mean ± SEM of $n = 22$ filaments from two cells.

applied by the cell to the substratum are diminished in Caldesmon-depleted cells. These defects do not arise from decreased myosin activation by phosphorylation, because myosin light chain (RLC) phosphorylation was unaltered in the Caldesmon knockout cells (Supplementary Fig. 7b–h). Moreover, FRAP analysis demonstrated the dynamics of NM-IIA heavy chains on ventral stress fibers were very similar in wild-type and Caldesmon knockout cells (Supplementary Fig. 7i–j).

## Caldesmon controls the force balance and myosin distribution within stress fibers

Because loss of Caldesmon leads to the abnormal organization of the stress fiber network, we next analyzed the effects of Caldesmon depletion on the distribution and dynamics of myosin II bundles. Importantly, by staining the cells with an antibody against NM-IIA heavy chain, we revealed an abnormal distribution of myosin filaments and filament stacks in the knockout cells plated on fibronectin-coated coverslips, as well as on crossbow micropatterns (Fig. 6a; Supplementary Fig. 7a). NM-IIA filaments displayed uniform distribution along transverse arcs and ventral stress fibers in wild-type cells, but the leading edges of Caldesmon knockout cells exhibited areas where NM-IIA stacks were clumped together, whereas other regions of transverse arcs were completely devoid of NM-IIA. Blind analysis of the images revealed that ~50% of the knockout cells displayed an irregular NM-IIA pattern along stress fibers, whereas the corresponding number in wild-type cells was <5% (Fig. 6b). This phenotype could be rescued by expressing full-length mCherry-Caldesmon in the knockout cells (Fig. 6c). Interestingly, the distributions of F-actin and different Tpm isoforms were still relatively uniform in the knockout cells (Supplementary Fig. 8c), whereas α-actinin distribution was somewhat affected by Caldesmon depletion (Supplementary Fig. 8a, b). More detailed 3D-SIM analysis revealed that in wild-type cells, α-actinin displayed complementary localization with Caldesmon with a distance of ~0.6 μm between the α-actinin peaks along stress fibers (Fig. 7a, b). On the other hand, the distances between α-actinin peaks along the stress fibers of Caldesmon knockout cells were more variable and typically shorter (Fig. 7a, c–e). Similarly, an actin filament pointed end capping protein of stress fibers, Tmod1, displayed a less regular pattern within stress fibers of Caldesmon knockout cells as compared to the wild-type cells (Supplementary Fig. 9). Of note, the actin filament barbed end capping protein (as visualized by eGFP–CP-β2 construct) did not localize to stress fibers in wild-type or Caldesmon knockout cells (Supplementary Fig. 10). These results indicate that also the organization of actin filaments within stress fibers is somewhat disturbed in the absence of Caldesmon.

To explore the role of Caldesmon in myosin distribution in more detail, we performed live-cell imaging of wild-type and Caldesmon knockout cells co-expressing RFP-actin and GFP-NM-IIA. In wild-type cells, myosin filaments assembled at the lamellipodia–lamellum interface, displayed consistent and uniform flow towards the cell center, and eventually fused with each other to form thicker NM-IIA stacks, as reported earlier[49–51]. In contrast, NM-IIA movements in

Caldesmon knockout cells were less uniform. Although the NM-IIA filaments appeared to assemble normally at the cell edge, they typically displayed abnormal lateral movements along transverse arcs and were further 'pulled' unequally towards opposite directions. These events resulted in NM-IIA filament accumulation at certain stress fiber regions as clumps of GFP-NM-IIA puncta, while the remaining stress fiber regions became devoid of NM-IIA filaments (Fig. 6d; Supplementary Fig. 11a, b; Supplementary Movies 8–10).

To dissect the functions of different Caldesmon domains in the regulation of NM-IIA distribution along stress fibers, we performed knockout-rescue experiments using Caldesmon deletion constructs (Fig. 8a). Blind analysis of NM-IIA distribution on transfected cells revealed that full-length mCherry-Caldesmon efficiently rescued the abnormal myosin distribution of knockout cells. In contrast, Caldesmon lacking the N-terminal (myosin-binding) region did not rescue the phenotype, as ~50% of cells displayed irregular NM-IIA distribution (Fig. 8b). Please note that this value is similar to the one quantified from the knockout cells (Fig. 6b). The construct lacking the central region of Caldesmon [Cald (1–200/374–531)] rescued the myosin phenotype to a large extent, with ~10% cells displaying irregular NM-IIA distribution. Surprisingly, the construct containing only the N-terminal (myosin-binding) region of Caldesmon [Cald (1–200)] partially rescued the myosin phenotype with ~30% of cells displaying irregular myosin distribution along stress fibers. However, many 'rescue' cells expressing this construct exhibited another unusual myosin phenotype, in which NM-IIA was largely dissociated from stress fibers, and accumulated in the perinuclear region of the cell (Fig. 8a, right panel). Finally, we carried out knockout-rescue experiments with a smooth muscle Caldesmon isoform (H-Cald), which is identical to the non-muscle L-Cald, apart from harboring ~250 residues longer linker region between the N- and C-terminal domains of the protein. Blind analysis of NM-IIA distribution revealed that H-Cald rescues the knockout phenotype quite well, but not to the same extent as the L-Cald (Supplementary Fig. 11c, d), indicating that the length of Caldesmon molecule may be fine-tuned to match the myosin II isoform expressed in the corresponding cell-type.

Collectively, these experiments revealed that Caldesmon prevents abnormal lateral 'sliding' of myosin II filaments along actin filaments and is therefore critical for maintaining the regular distribution of myosin II along stress fibers.

## Discussion

The mechanisms by which contractile actomyosin bundles are assembled in different cell types are relatively well understood, but how myosin II activity and force balance are controlled within contractile actomyosin bundles of smooth muscle and non-muscle cells has remained elusive. Here, we provide evidence that Caldesmon is critical for controlling myosin II distribution and force-balance within actomyosin bundles in non-muscle cells. We show that: (1) Caldesmon is an extended and highly dynamic component of contractile stress fibers, where it links myosin II filaments to specific tropomyosin-actin filaments, (2) Caldesmon does not function as a negative regulator of

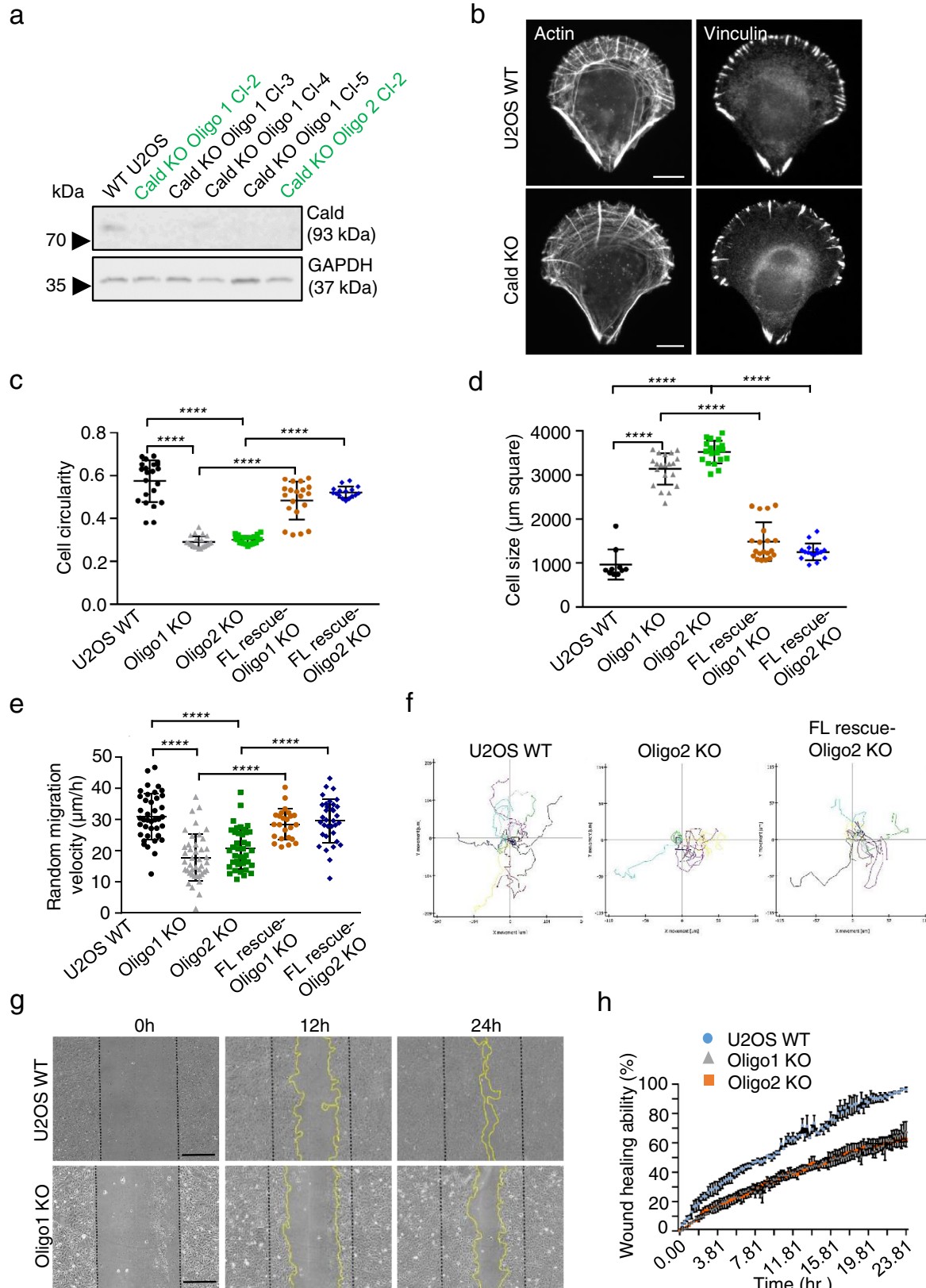

myosin II as previously proposed but instead maintains the regular spacing between individual myosin II filaments within stress fibers by preventing aberrant lateral 'sliding' of myosin filaments, and (3) Loss of Caldesmon leads to defects in the organization, contractility, and mechanosensing of the stress fiber networks, and to consequent problems in cell morphogenesis, migration, and invasion.

Our experiments provide evidence that the non-muscle Caldesmon splice variant (L-Caldesmon), is an extended molecule, whose N-terminal domain associates with the neck regions of myosin II molecules within NM-IIA filaments, while the C-terminal region of L-Caldesmon binds specific tropomyosin–actin filaments. These data are in agreement with previous electron microscopy work on purified

**Fig. 3 | Caldesmon knockout cells display defects in cell morphology and migration. a** Representative Western blot image showing Caldesmon protein levels in whole cell lysates of control and Caldesmon oligo1 and oligo2 knockout (KO) U2OS clones. The Caldesmon KO clones 2 generated with oligo1 and oligo2 Cas9 constructs (lanes 2 and 6, indicated in green) were chosen for further analysis. GAPDH was used as loading control. **b** Representative wide-field images of wild-type and Caldesmon oligo2 knockout U2OS cells grown on fibronectin-coated crossbow-shaped micropatterns and stained with vinculin-specific antibody and phalloidin. Caldesmon-depleted cells display less organized stress fiber networks compared to wild-type cells. Scale bars, 10 μm. **c** and **d** Analysis of cell circularity (panel **c**) and cell size (panel **d**) was carried out by high-content imaging of wild-type, oligo1 KO, and oligo2 KO cells, as well as oligo1 KO and oligo2 KO rescue cells expressing GFP-Caldesmon. Data represents mean ± SD $n = 58,600$ (wild-type), $n = 11,783$ (oligo1 KO), $n = 13,137$ (oligo2 KO), $n = 61,758$ (oligo1 KO-rescue) and $n = 49,997$ (oligo2 KO-rescue) cells analyzed from two repeats. Each data point in the graphs represents the mean value of a single well from a 96-well plate. Groups were compared with one-way ANOVA followed by Tukey's multiple comparison test. The p-values were adjusted for multiple comparisons. The error bars indicate 0.05 significance level with 95% CI. p-values: $****p = 0.0001$. **e** Graphical representation of random migration velocities obtained by high content imaging of wild-type ($n = 30$), oligo1 KO ($n = 37$), oligo2 KO ($n = 36$), oligo1 KO-rescue ($n = 24$) and oligo2 KO-rescue ($n = 34$) U2OS cells. Data represent mean ± SD from four experiments. **f** Representative random migration tracks of wild-type, oligo2 KO, and oligo2 KO-rescue U2OS cells demonstrating the directionality of cells. The values on X- and Y-axis denote the extent of migration by cells from their respective mean positions. **g** Representative examples of a wound healing assay illustrating the scratch area covered by wild-type and oligo1 KO cells at 0, 12, and 24 h time points. Scale bars, 400 μm. **h** Graphical analysis of percent (%) wound area covered by wild-type, oligo1 KO and oligo2 KO U2OS cells. The X-axis represents the time point h at which the % wound area covered (Y-axis) was calculated. The values represent mean ± SD from three independent experiments. Source data are provided as a Source Data file.

smooth muscle H-Caldesmon demonstrating that it is an elongated molecule in vitro[52], as well as with biochemical work providing evidence that the C-terminal region of H-Caldesmon binds actin and tropomyosin, while the N-terminal region of the protein interacts with myosin II[14–18,53–57]. Interestingly, our results propose that Caldesmon associates slightly better with actin filaments decorated by non-muscle tropomyosin isoforms Tpm1.6, Tpm2.1, and Tpm3.2, as compared to Tpm4.2-decorated or bare actin filaments. In this context, it is important to note that stress fibers are composed of functionally distinct actin filaments that are specified by different tropomyosin isoforms, and from these Tpm4.2 is most closely associated with myosin II function[9,58]. Caldesmon may thus preferentially interact with those actin filaments that do not form 'tracks' for myosin II movement. These data, together with our FRAP and domain depletion experiments, demonstrate that Caldesmon is a dynamic component that cross-links myosin II to tropomyosin-actin filaments (Fig. 8c).

Previous studies proposed that Caldesmon functions as a negative regulator of myosin II and/or it stabilizes actin filaments within contractile actomyosin bundles[13,27]. However, several lines of evidence from our work are not consistent with these earlier models. First, loss of Caldesmon results in decreased overall contractility and retrograde flow of the stress fiber network. Caldesmon depletion also does not increase the pre-stress within individual stress fibers, as would be expected if Caldesmon would function as a troponin-like negative regulator of myosin II activity. Second, the N-terminal myosin II-binding domain of Caldesmon is absolutely critical for the cellular function of the protein. On the other hand, the isolated C-terminal actin-binding region of Caldesmon still localizes to stress fibers, but cannot rescue the Caldesmon knockout phenotype, although previous biochemical studies proposed that this domain alone can regulate myosin II movement along actin filaments in a Calmodulin-regulated manner[13]. Finally, Caldesmon is a highly dynamic component of stress fibers and appears to preferentially associate with those tropomyosin–actin filaments that are not linked to myosin II movement. Thus, Caldesmon is not well-suited to function as a 'road block' for myosin II movement. It is also important to note that, since stress fibers are still present in Caldesmon-depleted cells, and because the N-terminal myosin II-binding domain is critical for its function in non-muscle cells, Caldesmon is also unlikely to function as an important stabilizer of actin filaments within stress fibers.

Our results reveal that Caldesmon prevents abnormal lateral 'sliding' of myosin II filaments along actomyosin bundles, and is thus critical in maintaining uniform spacing between individual myosin II filaments within stress fibers (Fig. 8d). Thus, rather than functioning as a negative regulator of myosin II similarly to troponin complex in striated muscles, we provide evidence that Caldesmon functionally resembles the M-band components of skeletal muscle sarcomere that are critical for maintaining the force-balance in muscles[6]. However, at the molecular level Caldesmon is very different from the M-band proteins, because Caldesmon links myosin II to tropomyosin–actin filaments, and is a highly dynamic protein within stress fibers. We speculate that this is due to the rapid dynamics of stress fibers compared to myofibrils. Thus, in stress fibers, there is a need for a more dynamic system that maintains uniform myosin II distribution and force balance. Our results, together with earlier studies[10,59–63], also suggest that the organization of actin filaments must be different between skeletal muscle sarcomeres and stress fibers of non-muscle cells. Unlike the highly regular length of actin filaments in muscle sarcomeres, we propose that tropomyosin-actin filaments, especially in transverse arcs, are of irregular length and can often extend beyond the adjacent α-actinin-rich Z-disk-like regions of stress fibers. This would allow myosin II filaments to move long distances laterally along transverse arcs, as observed in our live-imaging experiments. Moreover, loss of Caldesmon appears to result in the aberrant organization of actin filaments within stress fibers, together with laterally moving myosin II filaments. Although F-actin and tropomyosin distributions appear quite regular within the stress fibers of Caldesmon knockout cells, α-actinin appears to form larger aggregates in the absence of Caldesmon, and the distributions of α-actinin and Tmod1 along stress fibers are less regular in Caldesmon knockout cells. We speculate that these phenotypes are linked to the aberrant lateral movement of myosin II filaments because there are no reports indicating that Caldesmon would interact with α-actinin or Tmods.

In addition to linking myosin II filaments to Tpm-actin filaments, Caldesmon may also have more specific functions in cells. This is because Caldesmon also binds calmodulin, and this interaction inhibits the actin-binding of Caldesmon[19,20]. Thus, the stress fiber localization of Caldesmon may be controlled by $Ca^{2+}$/calmodulin. Moreover, as an elongated protein that links the necks of myosin II filaments to actin filaments, Caldesmon could, in principle, also function as a mechanosensor within stress fibers. However, our FRAP experiments revealed that Caldesmon is a highly dynamic component of stress fibers, and its dynamics are not affected by the inhibition of myosin. Thus, if Caldesmon also functions as a mechanosensor, it is most likely able to respond to forces only in a very short time scale.

In addition to its role in animal physiology, altered expression levels of Caldesmon were also linked to multiple cancers[13]. Our experiments demonstrate that in osteosarcoma cells, Caldesmon depletion results in decreased cell migration and invasion. However, in the complex tissue environment the loss of Caldesmon, and hence the defects in force-balance of actomyosin bundles, are expected to have different effects on cancer progression, depending on the type of cancer and tissue environment. In this context, it is important to note that based on the nuclear localization of YAP, our experiments suggest Caldesmon knockout cells are unable to sense substrate stiffness which is a property associated with cancer progression[64].

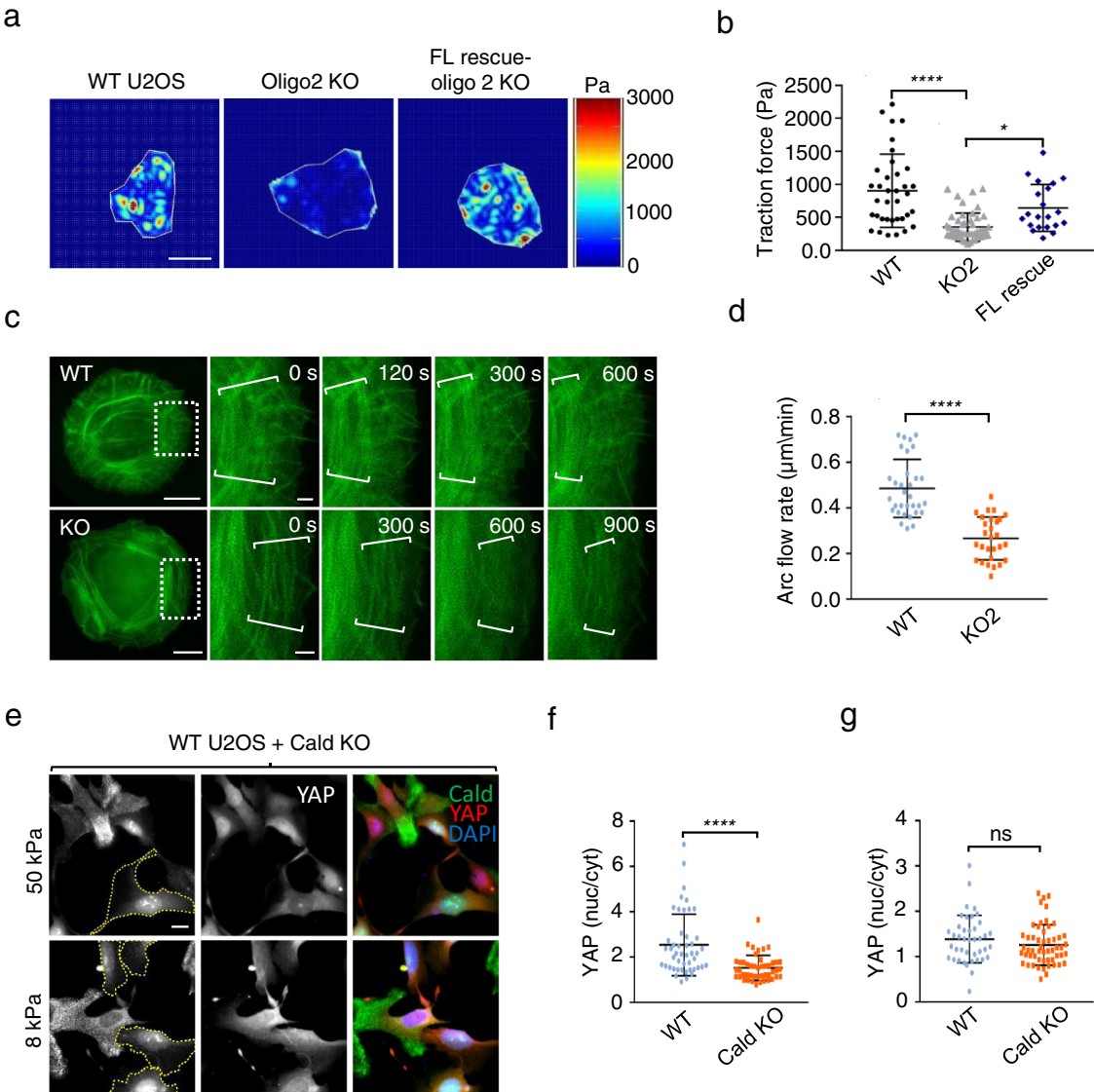

**Fig. 4 | Caldesmon is not a negative regulator of myosin II. a** Representative images of traction force maps of wild-type, oligo2 KO, and oligo2 KO-rescue U2OS cells grown on 26 kPa polyacrylamide covered with 488 fluorescent nanobeads and coated with collagen-1. The color scale indicates the stresses from 0 to 3000 Pascal (Pa). Scale bars, 20 μm. **b** Quantification of root means square (RMS) tractions of wild-type, oligo2 KO, and oligo2 KO-rescue U2OS cells. Data represents mean ± SD/ SEM from *n* = 36 (wild-type), *n* = 49 (oligo2 KO) and *n* = 21 (oligo2 KO- rescue) cells from three experiments (one way ANOVA followed by Brown–Forsythe's and Bartlett's test). *p*-values: **p* = 0.0133; *****p* = 0.0001. **c** Representative images of GFP-Lifeact expressing wild-type and oligo2 KO U2OS cells grown on fibronectin-coated circular micropatterns. White dashed boxes in the whole cell images (left) indicate the magnified regions (right). These illustrate the retrograde flow and condensation of the transverse arc network at indicated time points. Scale bars for whole cell images and magnified regions, 10 and 2 μm, respectively. **d** Retrograde flow rates of transverse arcs from wild-type (*n* = 32) and oligo2 KO (*n* = 28) filaments from four experiments. The data represents mean ± SD (Unpaired, nonparametric, two-tailed Mann–Whitney test). **e** Representative examples of wild-type and oligo2 KO (indicated by dotted lines) cells, mixed with each other, and grown on 8 and 50 kPa substrates. Cells were stained with antibodies against Caldesmon and YAP, as well as with DAPI to visualize nuclei. Scale bars, 20 μm. **f** Graphical representation of nucleus/cytoplasmic ratios of YAP in wild-type (*n* = 41) and oligo2 KO (*n* = 55) cells grown on 50 kPa matrix. **g** Graphical representation of nucleus/cytoplasmic ratios of YAP in wild-type (*n* = 47) and oligo2 KO (*n* = 46) cells grown on 8 kPa matrix. Data were analyzed using an unpaired, non-parametric Mann–Whitney test with two-tailed *p*-values representing mean ± SD in (**f**) and (**g**). Statistical significance for **d**, **f** and **g**: ns (*p* > 0.05; **p* < 0.0332); *****p* < 0.0001. Source data are provided as a Source Data file.

## Methods
### Cell culture and transfections
Human osteosarcoma (U2OS) cells (authenticated by ECACC through STR-profiling method to be the same origin as the original U2OS cell line; case number-13472) were maintained as previously described[11]. Briefly, cells were cultured in 4.5 g/L glucose containing DMEM (BE12-614F, LONZA), supplemented with 10% fetal bovine serum (10500-064, GIBCO) and Pen-Strep-Glutamine solution (10378016, GIBCO) in a humidified atmosphere at 37 °C, 5% $CO_2$ and

95% relative humidity. Cells were regularly tested for mycoplasma contamination using the Mycoalert™ Mycoplasma Detection Kit (LT07-418, LONZA). For overexpression studies, FuGENE® HD transfection reagent (E2312, Promega) was used as per the manufacturer's instructions with few standardizations using a 3.5:1 FuGENE HD/DNA ratio. Transfections were carried out for 24 h before live-cell imaging or before fixation with a 4% paraformaldehyde (PFA) solution. For rescue experiments, cells were transfected for 48 h.

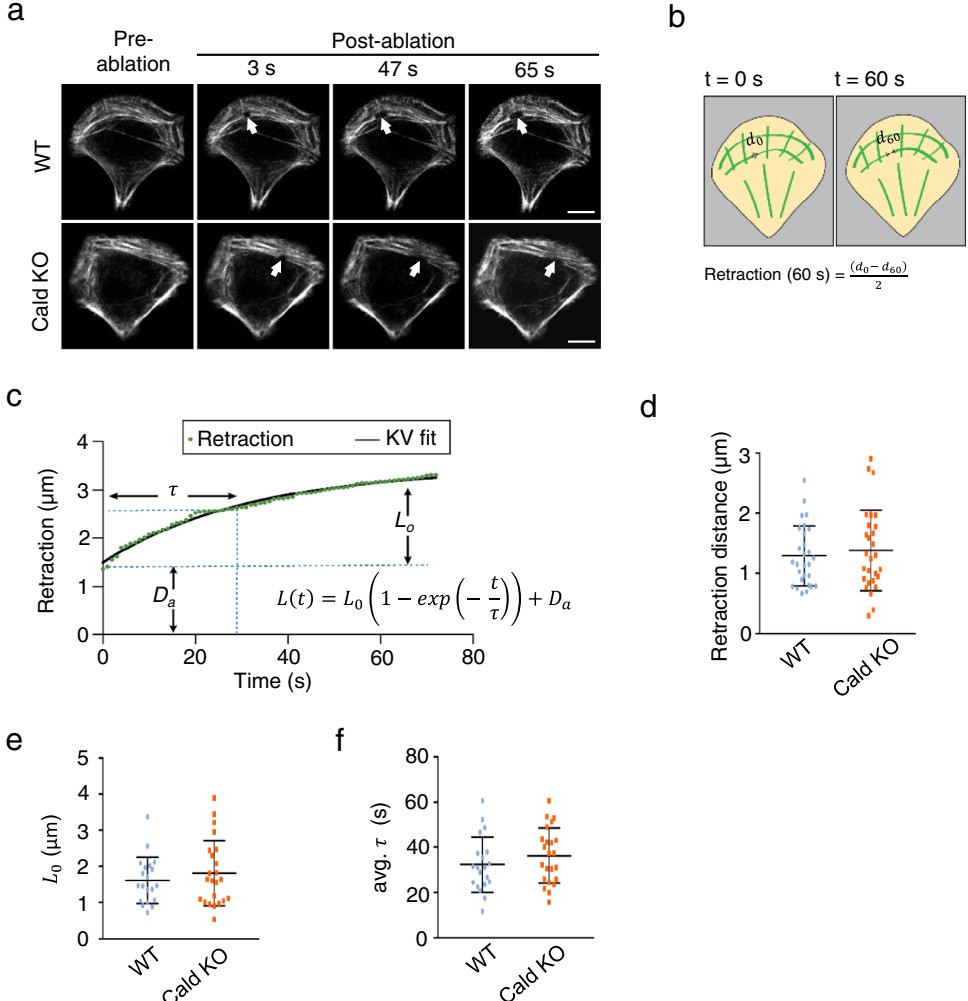

**Fig. 5 | Stress fiber mechanics in wild-type and Caldesmon knockout cells.**
**a** Single transverse arc ablation of RFP-LifeAct expressing U2OS wild-type and Caldesmon oligo2 KO cells on crossbow patterns. Top row is wild-type and bottom row is Caldesmon KO. Cells at 0 s illustrate the distribution of SFs pre-ablation, and cells at 3 s illustrate the distribution of SFs immediately after ablation. White arrows highlight the increase in retraction of the stress fiber post-ablation at 3, 45, and 65 s from the start of imaging. Scale bars = 10 μm. **b** Schematic of SF retraction measurement and model-independent SF retraction calculation. **c** Example of SF retraction tracked over time. The retraction distance is the half-distance between the severed ends at time $t$. The SF retraction is fitted to the Kelvin–Voigt model to extract plateau retraction distance ($L_0$), viscoelastic constant ($\tau$), and the length of the SF destroyed during photoablation ($D_a$) (see the "Methods" section). **d** Model-

independent SF retraction analysis of wild-type and Caldesmon KO U2OS cells. The distribution of the retraction at t = 60 s post-ablation of one cut end of the stress fiber for WT and KO cells. The data represents mean value ± SD from $n = 28$ (wild-type) and $n = 28$ (Caldesmon KO) cells from three experimental replications. Differences are non-significant (two-tailed, non-parametric Mann–Whitney $U$ test). **e** Viscoelastic time constant ($\tau$) and **f** Plateau retraction distance ($L_0$) for severed SFs in wild-type and Caldesmon KO U2OS cells. The data from panels **e** and **f** represent mean value ± SD from $n = 20$ (wild-type) and $n = 23$ (Caldesmon KO) cells from three experimental replicates. Differences are non-significant (two-tailed, non-parametric Mann–Whitney $U$ test). Non-KV retractions were excluded from the datasets of (**e**) and (**f**). Source data are provided as a Source Data file.

## Antibodies and chemicals
The following antibodies were used for western blotting (WB) and immunofluorescence (IF) microscopy: 1:10,000 (WB), 1:400 (IF) rabbit anti-Caldesmon (ab32330, Abcam, clone name-E89, lot number-GR1376472); 1:500 mouse anti-Caldesmon (sc-48427, clone name-F-1, lot number-#J1906, SCBT); 1:400 mouse anti-Vinculin (V9131, clone name-hVIN-1, lot number-079M4754V, Sigma-Aldrich); 1:1000 (WB), 1:800 (IF) rabbit anti-NM-IIA (909801, lot number- B292969, BioLegend); 1:200 (WB) mouse anti-RLC (M4401, clone name- MY-21, lot number- 046M818V, Sigma-Aldrich); 1:1000 rabbit anti-phospho-MLC (Thr18, Ser19) (3674, lot number-6, Cell Signaling Technology); 1:500 rabbit anti-α-actinin1 (A5044, Sigma-Aldrich); 1:500 rabbit anti-YAP (NB10-58358SS, lot number-B-5, Novus); 1:5,000 rabbit anti-GAPDH (G9545, lot number-#099M4801V, Sigma-Aldrich); 1:200 mouse anti-Tropomyosin 1 and 2 (T2780, clone name-TM311, lot number- #014M4782, Sigma-Aldrich). 1:200 mouse anti-Tropomyosin

2 and 4 (LC24) and 1:200 mouse anti-tropomyosin 3 (γ−9d) antibodies were kind gifts from Peter W. Gunning (University of New South Wales, Sydney, Australia). Alexa Fluor phalloidin 488 (1:400; A12379), phalloidin 568 (1:400; A1238) and phalloidin 647 (1:400; A22287) were used for filamentous-actin visualization; Alexa Fluor goat anti-mouse 488 (1:100; A11001, lot number-2015565), 568 (1:100; A11004, lot number-2124366), and 647 (1:100; A31571, lot number-1839633); Alexa Fluor goat anti-rabbit 488 (1:100; A11034, 1:100; A32733; lot number-UC279417) and 647 (1:100; lot number-UC279417, A21245) were purchased from Thermo Fisher Scientific for IF microscopy. Goat anti-rabbit HRP-conjugate (G21234), and goat anti-mouse HRP-conjugate (31430; both 1:5000) were purchased from Thermo Fisher Scientific for Western blotting experiments. 4′,6-diamidino-2-phenylindole, dihydrochloride (DAPI, 1: 1000, Thermo Fisher Scientific) was used for nuclear staining. Pageruler Plus prestained protein ladder (26619, Thermo Fisher Scientific) was used as a molecular weight marker in

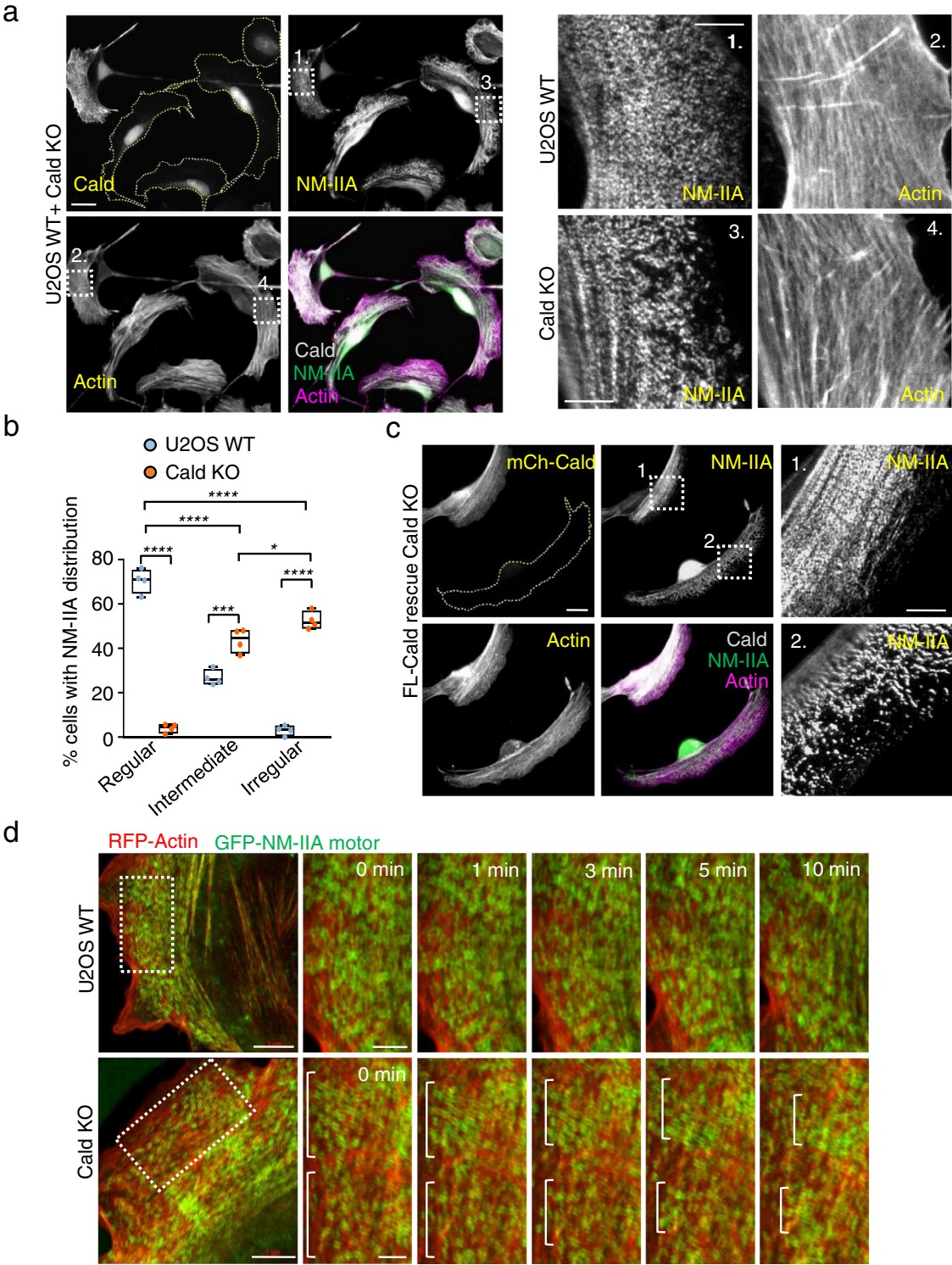

SDS−PAGE. *Para*-aminoblebbistatin (5 μM and 50 μM, DR-Am-89, Optopharma) was used for the NM-IIA inhibition. Fibronectin (10 mg/ml, 11080938001, Merck) was used for surface-coating in live and fixed-cell imaging studies. Treated coverglass was mounted using ProLong™ Glass Antifade Mountant (P36980, Invitrogen).

## DNA constructs

The following plasmids were used in this study: rat GFP-non-muscle-Caldesmon (gift from A. Bershadsky)[34]; mCherry-non-muscle Caldesmon was generated by amplifying Caldesmon-encoding fragment from rat non-muscle-Caldesmon construct. The fragment was sub-cloned further into pmCherry-C1 vector using the XhoI/BamHI restriction sites in the MCS region. For expression studies in U2OS cells, different domain deleted versions of Caldesmon namely−Cald(1-200), Cald(201-531), Cald(380-531), Cald(427-531), Cald(453-531), Cald(497-531), Cald(1-20), Cald(1-200/374-531), Cald(1-200/427-531), Cald(1-200/453-531), and Cald(1-200/497-531) were produced from mCherry-full-length Caldesmon by PCR, and sub-cloned into a relevant plasmid. For NM-IIA rescue experiments using smooth muscle-specific Caldesmon (H-Cald), pDONR221-Hs-H-Caldesmon (HsCD00746163) was procured from DNASU plasmid repository (Arizona State University, Arizona) and was sub-cloned further in pmCherry-C1 vector.

**Fig. 6 | Caldesmon maintains the regular distribution of myosin II along stress fibers. a** Representative examples of wild-type and oligo2 KO (indicated by dashed lines) U2OS cells mixed with each other, and stained with antibodies against Caldesmon, NM-IIA coiled-coil, and by phalloidin to visualize F-actin. Panels on the right are magnifications of the regions indicated with white boxes in the panels on the left, and they demonstrate regular patterns of NM-IIA (panel 1) along F-actin stress fibers (panel 2) in a wild-type cell. The distribution of NM-IIA filaments and filament stacks was irregular in Caldesmon knockout cells (panel 3), while the actin filaments still displayed uniform distribution also along those stress fiber regions devoid of NM-IIA (panel 4). Scale bars left 20 μm, and magnified images 5 μm. The images are representative of four experiments. **b** Blind analysis of the percent of wild-type and Caldesmon KO cells demonstrating regular, intermediate, and irregular distribution of NM-IIA filaments. Caldesmon depletion disturbs the normal distribution of NM-IIA filaments. The data represents wild-type ($n = 193$) and Caldesmon KO ($n = 267$) cells from four experimental repeats. The boxes indicate SD and the middle lines in the boxes are the mean values. The statistical significance was analyzed by two-way ANOVA with Tukey's multiple comparisons test. $p$-values:

$*p = 0.0433$; $***p = 0.0001$; $****p < 0.0001$. **c** Representative image of a Caldesmon KO cell (indicated by dashed line), and a knockout-rescue cell expressing full-length mCherry-Caldesmon. Cells were stained with an antibody against NM-IIA coiled-coil and with phalloidin. The magnified images (right) display regular NM-IIA filament distribution in the rescue cell expressing mCherry-Caldesmon (panel 1), whereas the non-transfected knockout cell (panel 2) displays irregular NM-IIA distribution along stress fibers. Scale bars, 15 and 5 μm in the whole cell and magnified images, respectively. **d** Representative time-lapse Zeiss Airyscan images of wild-type and Caldesmon KO cells expressing RFP-actin and GFP-NM-IIA. Magnified time frames (right) of the areas indicated by dashed boxes on left demonstrate the regular centripetal flow of NM-IIA stacks along transverse arcs in wild-type cells, whereas abnormal lateral 'sliding' of NM-IIA filaments occurs in Caldesmon knockout cells. This leads to NM-IIA accumulation in certain stress fiber regions (indicated by white brackets), while other stress fiber regions become largely devoid of NM-IIA. Scale bars, 5 and 1 μm in left and in magnified time-frames, respectively. Source data are provided as a Source Data file.

For biochemistry, FL-Cald, Cald(1-200), Cald(201-531), and Cald(380-531) were tagged with His10pSUMO in pBR322 vector. All primers used for cloning purposes are listed in Supplementary Table 1. mCherry and mRuby-labeled and non-labelled constructs of tropomyosin 1.6, 2.1, 3.2, and 4.2 used in vivo and in vitro were from the previous studies[65]. pSpCas9 (BB)−2A-GFP vector was a gift from F. Zhang (Addgene plasmid #48138). CMV-GFP-NMHC-IIA was a gift from Robert Adelstein (Addgene plasmid #1347)[66] and NM-IIA-GFP was a gift from Matthew Krummel (Addgene plasmid #38297). GFP-lifeact was a kind gift from Maria Vartiainen (University of Helsinki, Finland) and pEGFP-C1-F-tractin was a gift from Dyche Mullins (Addgene plasmid #58473). eGFP-CP-β2 was a kind gift from D. Schafer (University of Virginia, Chalottesville)[67]. pEGFPN1-HsTmod1_wt was available from a previous study[11].

### Knockout cell lines generation and confirmation

Caldesmon knock-out cell lines were generated by CRISPR/Cas9 method as described previously[68–71]. Guide sequences 5′-CACCGG CCGTTCCTGTCGGGCTCGG and 5′-CACCGGGGACAGGTGACCGACCA GG specific to exon 1 of Caldesmon gene having highest on-target efficacy scores were selected based on CRISPR design tool and were cloned into pSpCas9 (BB)−2A-GFP vector. The constructs were transfected in U2OS cells and positive cells were sorted individually with FACSAria II cell sorter (BD) supplied with BD FACSDiVa Software v 8.0 into a 96-well plate containing DMEM media supplemented with 20% FBS and 25 mM HEPES. Caldesmon deletion in these knockout cell lines was confirmed on protein level by Western blotting, and the underlying genomic alteration was identified by Sanger Sequencing (Eurofins genomics sequencing service) and next generation sequencing (NGS) (Illumina MiSeq.) that was performed at the DNA Sequencing and Genomics Laboratory (BIDGEN) laboratory (Institute of Biotechnology, University of Helsinki, Finland).

### Protein purification

Full-length Caldesmon and its truncated versions cloned in pBR322 were expressed in *E. coli* BL-21 (DE3) pLys-S cells and grown in 2X lysogeny broth containing Kanamycin (25 μg/ml) and Chloramphenicol (35 μg/ml) at 37 °C until the $OD_{600}$ reached 0.8–1.0. Protein expression was induced with 0.5 mM IPTG for 24 h at R.T. Cells were harvested by centrifugation (4000×$g$, 4 °C, 15 min), re-suspended in Ni-binding buffer (50 mM Tris−HCl pH-7.5, 300 mM NaCl, 10 mM Imidazole pH-8.0, 1 mM DTT and 1X protease inhibitor cocktail (06538282001, Roche Applied Science) and lysed thoroughly by sonication. Cell lysates were mixed with Ni-NTA agarose beads (30230, Qiagen) in a Nickel binding buffer at 4 °C for 1 h. The beads were washed with four column volumes (CV) of binding buffer, 6 CV of 1 M NaCl solution followed by elution with Ni-elution buffer (50 mM Tris−HCl pH-7.5, 300 mM NaCl, 250 mM Imidazole, 1 mM DTT). SUMO tag was removed by SENP protease overnight in dialysis buffer (50 mM

Tris−HCl pH 7.5, 300 mM NaCl, 1 mM DTT) and proteins were further purified on gel filtration column pre-equilibrated with 20 mM HEPES pH-8.0, 50 mM NaCl. Fractions containing Caldesmon proteins were pooled together, concentrated, loaded in ion exchange HiTrap Q FF columns (17-5053-05, GE Healthcare Bio-Sciences AB), washed with 10 mM Tris−HCl pH 7.5, 10 mM NaCl, and eluted using 10 mM−1 M NaCl salt gradient. Finally, the fractions containing pure protein were pooled together, concentrated with Amicon® Ultra-15 centrifugal filters (30,000 NMWL-UFC903024; 10,000 NMWL- UFC901096, Merck Millipore Ltd.), and protein concentration measured with nanodrop using an extinction coefficient of (protparam), snap frozen in liquid nitrogen and stored at −80 °C.

The plasmids for the structural and biochemical studies of non-tagged human tropomyosins Tpm1.6, 1.7, 2.1, 3.1, 3.2, and 4.2 were from earlier study[65]. Non-tagged tropomyosins were purified as per a published method[72] with some essential modifications. Briefly, non-tagged tropomyosins were expressed in *E.coli* BL21-DE3 cells in 1 l LB broth (2X) containing ampicillin (100 μg/ml) and were grown until the $O.D_{600}$ reached 0.8–1.0. The protein expression was induced with 1 mM IPTG at 37 °C for 3 h. The induced cells were harvested by centrifugation, resuspended in lysis buffer (20 mM $Na_2PO_4$, 500 mM NaCl, 5 mM $MgCl_2$, 1 mM DTT, and protease inhibitor cocktail), and were lysed thoroughly by sonicating (2 min, 50% amplitude, 30 s 4× on/off interval) until the bacterial suspension turns sufficiently transparent. Further, the cell lysate was heated in a water bath set at 80 °C for 8 min. The heat-inactivated cell lysates were subsequently cooled to room temperature in a water bath. The supernatant containing Tpms was cleared from denatured proteins and cell debris by centrifugation at 47,810×$g$, 4 °C for 45 min. The supernatant was collected in a fresh tube and the Tpms were acid precipitated to pH 4.7 by adding drop-by-drop 2 M HCl while mixing. The precipitate containing Tpms was centrifuged at 3900×$g$, 4 °C for 12 min, and the pellet was resuspended in resuspension buffer (100 mM Tris−Cl pH 7.5, 500 mM NaCl, 5 mM $MgCl_2$, 1 mM DTT, 1 mM $NaN_3$). In order to resuspend the pellet completely, the buffer pH was re-adjusted to 7.0 using 1 M NaOH. The acid precipitation was repeated two more times to remove impurities and obtain a white pellet. The white pellet was re-suspended in pellet resuspension buffer and was filtered using a 0.22 μm membrane filter. Dialysis was performed overnight in dialysis buffer (20 mM Tris−Cl pH 7.5, 500 mM NaCl, 0.5 mM DTT) using 10 kDa molecular cutoff membrane (#88243, SnakeSkin™ Dialysis Tubing, 10 MWCO, Thermo Scientific). The next day, a 4 ml SP QFF anion exchange column was equilibrated with 4 column volumes of anion exchange wash buffer (20 mM Tris−Cl pH 7.5, 10 mM NaCl, 0.5 mM DTT) and the supernatant was loaded onto the column at speed of 2 ml/min. The column was washed with four column volumes of wash buffer. The column was further equilibrated with anion exchange elution buffer (20 mM Tris, pH 7.5, 1 M NaCl, 0.5 mM DTT) and Tpm was collected in the flow

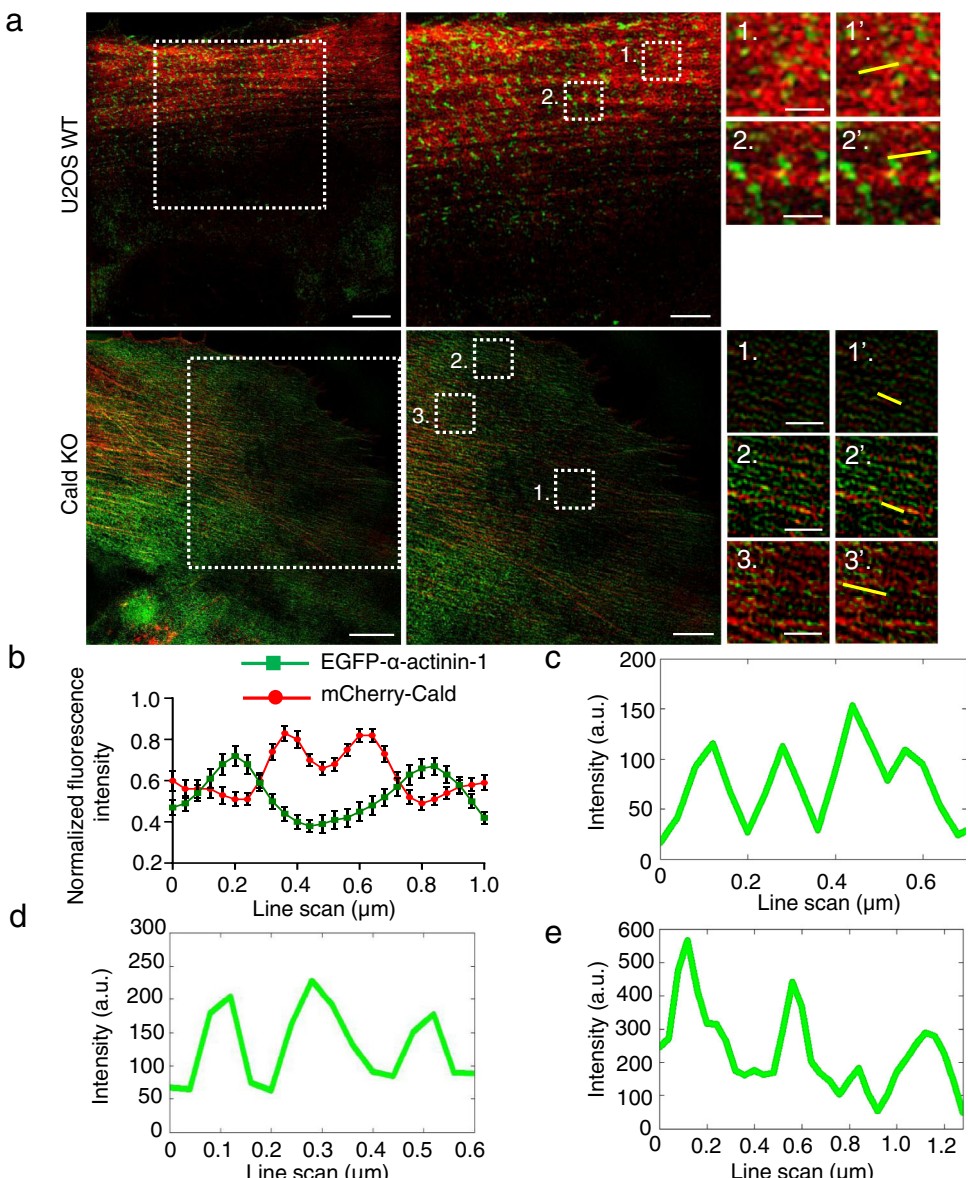

**Fig. 7 | Effects of Caldesmon depletion on α-actinin localization along stress fibers. a** Representative SIM image of a wild-type U2OS cell (top row) expressing GFP-α-actinin (green) and mCherry-Caldesmon (red). White boxes represent the magnified regions. The thin yellow lines in magnified images 1' and 2' represent examples of stress fiber regions that were used for the line plot analysis of α-actinin/Caldesmon co-localization. The bottom row shows representative SIM images from a Caldesmon KO cell expressing GFP-α-actinin (green) and stained with phalloidin for F-actin (red). The magnified regions 1, 2, and 3 represent the localization of α-actinin in F-actin-rich stress fibers in a Caldesmon KO cell, and the thin yellow lines in 1', 2' and 3' show regions that were used for the line plot analysis. **b** Line plot analysis of the co-localization pattern of α-actinin and Caldesmon in wild-type U2OS cells. Line scans = mean ± SEM of $n$ = 29 filaments from 3 cells. **c–e** Representative line scans of selected regions (yellow lines in 1', 2', and 3') from a Caldesmon KO cell demonstrating irregular distribution of α-actinin rich regions within stress fibers. Scale bars, 5, 3, and 1 μm for the left, middle, and right panels, respectively.

through. Samples from peak fractions were analyzed by SDS−PAGE and fractions were pooled together. The pooled fractions were diluted four times with hydroxyapatite wash buffer (10 mM $Na_2PO_4$ pH 7.0, 1 M NaCl, 0.5 mM DTT) and loaded on an equilibrated hydroxyapatite column at 0.5 ml/min speed. The column was washed with four column volumes of hydroxyapatite wash buffer and the protein was eluted in a 10−240 mM phosphate gradient against hydroxyapatite elution buffer (240 mM $Na_2PO_4$ pH 7.0, 1 M NaCl, 0.5 mM DTT). The eluted fractions were analyzed by SDS−PAGE and the fractions containing Tpms were pooled together. As the elution volume was large, we utilized an acidification procedure to concentrate the protein. The protein pellet was resuspended in dialysis buffer (20 mM Tris−Cl pH 7.0, 100 mM KCl, 5 mM $MgCl_2$, 0.5 mM DTT, 0.02% $NaN_3$), protein concentration

was determined with nanodrop, the protein sample was aliquoted, flash-frozen, and stored at −80 °C.

### Western blotting

A total of $2 \times 10^6$ cells were plated on 35 mm dishes 18−24 h before harvesting. Cells were washed twice with ice-cold PBS solution and the cells were harvested on ice in PBS containing protease and phosphatase inhibitor cocktails using a cell scrapper. Pelleted cells were lysed using 4× Laemmli buffer + β-ME. Whole-cell extracts were run on 4−20% gradient SDS−PAGE gels (4561094, Biorad) followed by transfer onto 0.2 μm nitrocellulose membrane (1704158, Trans-blot turbo transfer pack, Biorad) using Trans-Blot® Turbo™ transfer system (1704150, Biorad). Blots were blocked for 1 h at RT in 2.5% BSA in PBS-T

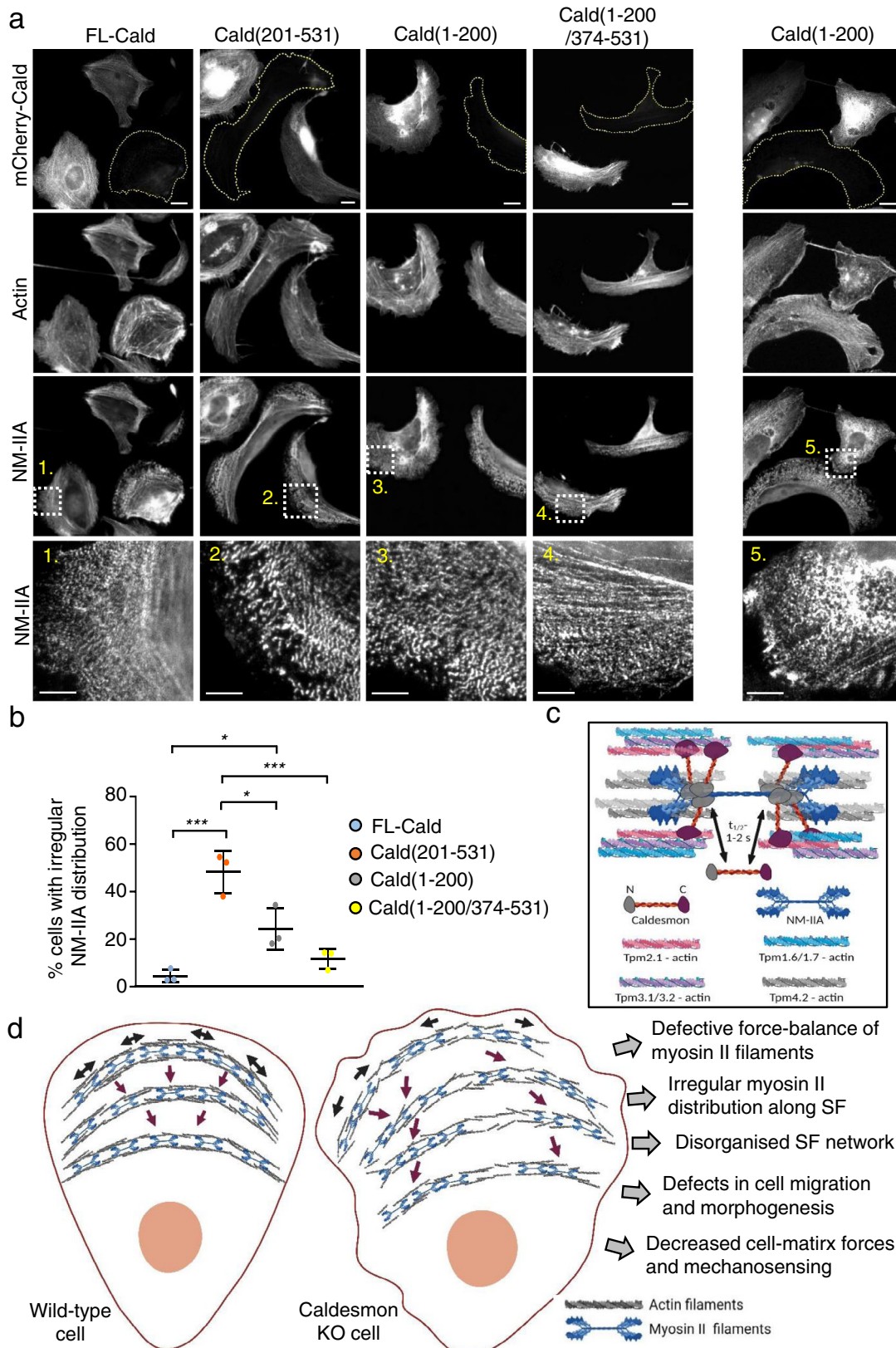

(0.05% Tween-20) followed by probing with primary antibody against phospho- and total-proteins on overnight shaking at 4 °C. The blots were washed three times in 0.05% TBS-T and incubated with respective HRP-conjugate secondary antibodies for 1 h at RT. Finally, the blots were washed and protein bands were detected using WesternBright™ ECL-spray (K-12049-D50, Advansta) in the ChemiDoc XRS+ System

(1708265, Biorad). Protein bands were quantified using Image Lab™ (1709690, Biorad) and Fiji ImageJ 1.53c [https://imagej.nih.gov/ij/] software. GAPDH was used to normalize expressions of other proteins and phospho-expression was normalized with respective total-protein levels. Densitometry analysis of Western blots and the uncropped Western blots are provided in the Source Data files.

**Fig. 8 | Function of Caldesmon and its different domains in keeping myosin II filaments on track. a** Representative examples of Caldesmon oligo2 KO U2OS cells transfected with mCherry-full-length-Caldesmon, mCherry-Cald (1–200), mCherry-Cald (201–531), and mCherry-Cald (1–200/374–531) constructs. NM-IIA and F-actin were visualized by an antibody against NM-IIA coiled-coil and by phalloidin, respectively. Dashed lines highlight non-transfected Caldesmon KO cells, whereas Caldesmon-transfected cells can be distinguished by mCherry-expression (first row). Magnified images at the bottom row, corresponding to the white dashed boxes above, highlight the distribution of NM-IIA filaments in respective cells. Scale bars, 15 and 5 μm in the whole cell and magnified images, respectively. Additionally, an example of a Caldesmon knockout cell expressing mCherry-Cald (1–200) and displaying abnormal perinuclear accumulation of NM-IIA is shown in the column on right. **b** Blind analysis of irregular NM-IIA filament distribution along stress fibers in Caldesmon knockout U2OS cells expressing different Caldesmon constructs depicted in panel (**a**). The data represent mean ± SD (1-way ANOVA followed by Tukey's multiple comparisons test) for full-length Cald (n = 105), Cald (1–200) (n = 135), Cald (201-531) (n = 160), and Cald (1–200/374–531) (n = 121) cells from

three experiments. Statistical significance: ns (p = 0.1234); *(p = 0.0293: FL-Cald vs. Cald(1-200), p = 0.0105: Cald(201-531) vs. Cald (1-200)); ***(p = 0.0002: FL-Cald vs. Cald(201-531), p = 0.0008: Cald(201-531) vs. Cald(1-200/374-531)). **c** Caldesmon is an extended protein and a highly dynamic component of stress fibers with a half-life of 1–2 s. Caldesmon interlinks myosin II to tropomyosin-decorated actin filaments by associating through its N-terminal domain with the 'neck-region' of myosin II, and with tropomyosin-actin filaments through its C-terminal domain. We further propose that myosin II motor domains associate with Tpm4.2 decorated-actin filaments (refs. 9,64), whereas Caldesmon prefers to bind actin filaments decorated by other non-muscle tropomyosin isoforms. **d** In wild-type cells (left), myosin II filaments display regular spacing within transverse arcs. In the absence of Caldesmon (right), myosin II filaments exhibit aberrant lateral sliding along actin filaments, and this results in irregular distribution of myosin II filaments in stress fibers. Consequently, the organization and force-balance of the stress fiber network become disturbed, and this further leads to defects in cell morphogenesis, migration, invasion, and mechanosensing. Source data are provided as a Source Data file.

## F-actin co-sedimentation assay

We performed actin co-sedimentation assays as previously reported[73] with few modifications. Non-muscle actin (β/γ-actin from human platelets) was purchased from Cytoskeleton Inc. and used as per the manufacturer's instructions. Different amounts of either β/γ-actin or β/γ-actin and non-tagged tropomyosin 1.6, 2.1, 3.2, and 4.2 were mixed in presence of G-buffer (5 mM HEPES pH 7.4, 0.2 mM CaCl₂, 0.2 mM DTT and 0.2 mM ATP). In order to saturate actin with tropomyosins, we mixed actin and tropomyosins in 4:1 ratio. Caldesmon concentration was maintained at 1 μM for all experiments. Incubations were done at RT. Polymerization was carried out for 30 min using F-buffer (20 mM HEPES pH 7.4, 100 mM KCl, 5 mM MgCl₂, 0.2 mM EGTA, 1 mM DTT and 0.2 mM ATP). 1 μM of Caldesmon was added to the polymerized actin/actin: Tpm mix and was incubated for 30 min. The complex was centrifuged at 157,000×g for 60 min at 10 °C (TLA100 rotor, Beckman Optima MAX Ultracentrifuge). Supernatant and pellet fractions were prepared for SDS–PAGE analysis by adding Laemmli buffer. The protein bands were separated on 4–20% gradient SDS–PAGE gels (Mini-PROTEAN TGX Precast Gels, Bio-Rad Laboratories Inc.) and stained using 1% Coomassie staining solution (25% isopropanol, 10% glacial acetic acid) (Coomassie brilliant blue R-250, 1610400, Biorad Laboratories, Inc.). The intensities of Caldesmon bands were quantified with QuantityOne program (Bio-Rad). The graph was plotted as the concentration of Caldesmon in pellet (μM) on Y-axis vs. actin/actin:Tpm1.6/actin:Tpm3.2 (μM) on X-axis.

## Immunofluorescence staining

Immunostaining (both for wide-field and 3D-SIM) was performed as reported previously[74]. Briefly, cells were seeded on 10 mg/ml fibronectin-coated coverglass and allowed to spread for 6 h. The old media containing floating cells was removed and coverslips were washed twice with PBS. Cells were fixed using 4% paraformaldehyde in PBS for 15 min at RT, followed by three washes with PBS-T (0.02% Tween-20). Cells were permeabilized with 0.1% Triton X-100 in PBS (10 min, RT) and blocked with 1% BSA in a humidified chamber for 1 h at RT. Cells were immunostained with primary antibodies for 1 h at RT, washed three times with PBS-T, and incubated with Alexa Fluor-conjugated secondary antibodies for 1 h at RT in dark. When relevant, F-actin and nuclei were visualized using phalloidin and DAPI. Finally, the coverslips were washed with PBS-T three times, with PBS once, and mounted using an anti-fade mounting reagent.

## Wide-field imaging

Stained cells were visualized at RT on a Leica DM5000B and DM6000B upright microscope with ×63/1.40-0.60 HCX PL APO oil objective, equipped with Semrock bright line DAPI, FITC/GFP, mCherry/TRITC, and Cy5 filters. Wide-field fluorescence images were captured using

Hamamatsu Orca-Flash4.0 V2 sCMOS camera with image resolution 2048 × 2048 pixels. The images were acquired using LAS X 3.4.2 software and processed using Fiji ImageJ 1.53c software.

## Confocal imaging

Wild-type and Caldesmon KO cells were seeded on Softview™ 35 (10 mm Glass Bottom), Easy Coat™ dishes of 8 kPa (SV3520-EC-PK, Cell guidance systems) and 50 kPa (SV3520-EC-50 PK) substrate stiffness. Confocal imaging was performed at RT on Andor Dragonfly confocal microscope using ×10 Plan Fluor DL/0.3 NA/15.2 WD and ×20 S Plan Fluor ELWD/0.45 NA/8.2–6.9 WD objectives, equipped with 445/521/594/685/809 filters for imaging. Images were captured using Andor iXon 888 U3 EMCCD camera and visualized using Fusion 2.0 iXon SRRF-Stream real-time super-resolution software.

## Super-resolution imaging

For SIM imaging, precision coverslips with 170 μm thickness were used. The imaging was performed at RT on a GE Deltavision OMX super-resolution upright microscope with a Plan-Apochromat N ×60/1.42 Oil immersion objective, equipped with 405, 488, 568, 640 nm diode lasers and sCMOS detector camera. Images were acquired using AquireSR 4.4 software, reconstructed, and aligned using SoftWoRx 7.0 software as described before[37]. GE immersion oil calculator was used to calculate the refractive index of oil to be used while imaging images stained with two or more fluorophores. SIM images were constructed from average intensity projections of Z-stacks using Stacks/Z-project and analyzed using 3DimageJ plugins in imageJ 1.53c software. For the line-scan analyses, mCherry-Caldesmon doublets (with a distance of ~0.2 μm between the two peaks) were chosen and manually centered on the analyzed ROI. The fluorescence intensities of mCherry-Caldesmon and the corresponding second protein were then measured along the ROI.

## High content imaging

For the analysis of cell shape and size, wild-type and Caldesmon KO untransfected/mCherry-Caldesmon transfected U2OS cells were plated in a fibronectin-coated 96-well plate (655180, Greiner) and incubated for 6 h. The plated cells were counter-stained with DAPI, and phalloidin 488. Cell imaging was done at RT using ImageXpress® Nano Automated Imaging System (Molecular Devices) with ×20/0.45S Plan Fluor ELWD, WD 8.2–6.9 mm water immersion objective, and 409, 495, 593, and 660 nm dichroic filters. The images were processed in CellProfiler 3.1.8 automated image analysis software. A two-color and three-color pipeline was designed for machine learning and to process untransfected wild-type, Cald KO, and transfected Cald KO U2OS cells. DAPI served as a primary object filter for cell segmentation. Object aggregates and smaller artifacts were discarded automatically based

on intensities by setting pre-filters like threshold and correction factor. After the primary screening, secondary objects positive for phalloidin and GFP-Caldesmon were identified. Post-regularization, the intensities of objects were measured and filtered and the shape-size of the positive objects was calculated automatically.

## Live cell imaging

Live cell imaging was performed as previously reported[37,42]. Wild-type and Caldesmon KO U2OS cells expressing RFP-actin and NM-IIA-GFP constructs were plated on fibronectin-coated 35 mm glass bottom dishes (81158, Ibidi) after 24 h of transfection. The cells were incubated for attachment for 2 h. Time-lapse live-cell imaging of RFP-actin and NM-IIA-GFP myosin motor accumulation and retrograde flow was performed using Zeiss LSM 880 upright confocal microscope combined with AiryScan (Fast) detector at 37 °C, with a Plan-Apochromat ×63/1.4 NA/0.19 WD Oil immersion objective. Time-lapse images were captured using Zeiss Zen 2.0 software for 30 min at 10 s intervals. Deconvolution was done using ZEN black software. The retrograde flow of NM-IIA in wild-type and Cald KO cells was manually analyzed in Fiji imageJ 1.53c. Wild-type and Caldesmon KO U2OS cells were transfected with GFP-tractin for 24 h, and re-plated before imaging for 2 h on fibronectin pre-coated CYTOOchips™ circular micropatterns 35 mm coverglass (10-900-10-18, Cytoo) attached with a CYTOO-chamber™ (30-010, Cytoo). The micropatterns were washed twice with pre-warmed PBS and time-lapse images were acquired with 3I Marianas imaging system (3I intelligent Imaging Innovations)−a fully motorized Zeiss inverted spinning disk confocal microscope (Zeiss Axio Observer Z1). The microscope is equipped with WC-Apochromat ×63/1.2 NA/Corr. WD = 0.28 M27 water objective and Yokogawa CSU-X1 M1 5000 rpm confocal scanner with Andor Neo sCMOS camera. The microscope has a 405/488/561/640 dichroic filter and a brick $CO_2$ mixer with active temperature and humidity control. Images were acquired with Slidebook 6.0.15 software. The flow of transverse arcs in wild-type and Caldesmon KO U2OS cells were tracked manually in Fiji imageJ1.53c software and the velocity of retrograde flow of transverse arcs was calculated.

## Fluorescence recovery after photobleaching (FRAP)

FRAP experiments were performed as described previously[42]. To measure the dynamics of actin, Caldesmon, and NM-IIA, wild-type, and Caldesmon KO U2OS cells were transfected with GFP-actin, GFP-NM-IIA, and GFP-Caldesmon constructs, incubated for 24 h before seeding them on fibronectin-coated glass-bottomed 35 mm imaging dishes (81158, Ibidi) for 2 h. The old media containing un-attached cells was replaced with fresh media before live-cell imaging on 3I Marianas imaging system (3I Intelligent Imaging Innovations)-Yokogawa CSU-X1 M1 5000 rpm spinning disk confocal microscope at 37 °C and $CO_2$ control using ×63/1.2 W C-Apochromat Corr WD = 0.28 M27 water objective and Andor Neo sCMOS camera. A bleaching area was marked on ventral stress fibers having uniform length and thickness across different transfected groups. Before bleach (100% intensity with 150 mW−488 nm laser power for 1 × 1 ms), five pre-bleach images were captured, followed by bleaching and post-bleach imaging to monitor photo-recovery at 50 × 200 ms and then at 300 × 1 s, respectively. To measure the photo-recovery, the intensity of the bleached area was normalized to the neighboring non-bleached area on the same stress fiber. Normalized values of different groups obtained from experimental repeats were used to calculate the mean recovery and variation across replicates for actin, Caldesmon, and NM-IIA in wild-type and Caldesmon KO U2OS cells.

## Traction force microscopy (TFM)

**Substrate preparation.** The polyacrylamide (PAA) gel substrates with 26 kPa stiffness (Young's modulus) were prepared as described previously[75,76] on glass-bottomed 35 mm imaging dishes (CELLView™, Greiner bio-one). The gels were covered with fluorescent 488-nano beads (diameter 200 nm) and after that further coated with collagen-1. Prior to cell plating, gels were incubated with warmed 10% FBS and supplement-containing DMEM media for 1 h.

**Cell transfections and live-cell imaging.** Wild-type and Caldesmon KO U2OS cells transfected with GFP-Lifeact construct were incubated for 24 h. Wild-type, untransfected and mCherry-FL-Caldesmon transfected U2OS cells were cultured on the collagen-1-coated 35 mm imaging dishes for 2 h at 37 °C. After a brief wash, we imaged the attached cells and the fluorescent beads beneath the cells using the 3I Marianas imaging system (3I Intelligent Imaging Innovations) by bright field and epi-fluorescence settings before and after cell-detachments by trypsinization (microscope specifications and imaging setup as described before in live-cell imaging section). Displacement maps were received by comparing the bead images before and after trypsinization. Traction fields were computed from the displacement data by utilizing the principle of Fourier transform traction cytometry as described before[77,78]. Cell borders were manually tracked and analyses of different groups were performed as blind analyses.

## Laser nanosurgery

Laser ablation studies were performed as previously described[47,48]. Briefly, U2OS cells were transduced with RFP-Lifeact and cultured on either unpatterned surfaces, rectangular frame patterns, or crossbow patterns. Micropatterns were made as previously described[46–48] Briefly, coverslips were plasma-treated and then incubated with 50 μg/ml poly-L-lysine conjugated to polyethylene glycol (PLL-g-PEG; SuSoS) in 10 mM HEPES, pH 7.4, for 1 h at room temperature, then rinsed with PBS and deionized water. To pattern the surface, coverslips were placed under a quartz-chrome photomask with micropattern features (Front Range Photomask, features were designed using AutoCAD). The mask and coverslips were then illuminated using 180 nm UV light (Jelight) for 15 min and then rinsed with PBS. Both patterned and unpatterned coverslips were incubated with 20 μg/ml fibronectin (EMD Millipore) in 100 mM bicarbonate solution, pH 8.5, overnight at 4 °C, then rinsed with PBS. U2OS cells were seeded 3000–5000 cells/cm² and incubated for 4–6 h at 37 °C to allow for attachment and spreading. Imaging and ablation were performed with a Zeiss LSM 780 confocal microscope system equipped with a Spectra-Physics Mai-tai laser for multiphoton imaging and ablation. Single stress fibers were then photo-irradiated with the femtosecond laser to induce an incision. The retraction was tracked every 1 s for 75 s and fit to a Kelvin−Voigt single-exponential model to extract a viscoelastic time constant ($τ$) and plateau retraction distance ($L_0$). Retraction curves were excluded from the $τ$ and $L_0$ datasets if they did not meet all of the following criteria for the fitted KV equation: (1) adjusted $R^2 > 0.9$, 2) $τ < 0.8 *$ imaging window ($τ < 60$), and (3) sum squared errors of prediction < 10. The model-independent analysis of retraction was calculated by subtracting the distance between the severed ends at 60 s and at 0 s and dividing by 2.

## Random migration assay

Random cell migration velocities were studied as reported previously[70]. Wild-type and Caldesmon KO cells untransfected or GFP-FL-Caldesmon transfected were cultured for 2 h on a 12-well plate coated with fibronectin before imaging. Live-cell imaging was performed using a Cell-IQ imaging system by Nikon Plan Fluor ×10/0.30 Ph1 DL objective (fluorescent and phase-contrast mode) and the time-lapse images of cells were captured with Qimaging Retiga EXi camera (pixel no. ($W × H$): 1392 × 1040) with 2 × 2 binning at 10 min intervals for 24 h. The system utilizes Cell-IQ Imagen 4.1.0 and Cell-IQ Analyser 4.4.0 software for cell imaging and analysis. The fluorescence and phase-contrast images were stitched and combined together to differentiate between transfected and non-transfected cells. To assess the

migration velocity in Caldesmon KO-rescue cells, only GFP-Caldesmon-transfected cells were analyzed, while non-transfected cells determined the Caldesmon KO cell migration velocity. Cells colliding with each other or undergoing division were omitted during the analysis and the measured migration velocities across groups were used to plot scatter box plots.

## Wound healing assay

Wound-healing property of wild-type and Caldesmon KO U2OS cells was assessed by wound healing assay. $5 \times 10^5$ cells were seeded in a 12-well culture plate and incubated for 18–24 h to form a uniform cell-monolayer. Scratch was made with 200 μl micropipette tip in the mid-area of every well, media replaced with 10% FBS + 10 mM HEPES containing-DMEM media and time-lapse live cell imaging performed using Cell-IQ imaging system by Nikon Plan Fluor 10x/0.30 Ph1 DL objective (phase-contrast mode). Images were captured with Q imaging Retiga EXi camera (pixel no. $(W \times H)$: $1392 \times 1040$) with $2 \times 2$ binning at 10 min intervals for 24 h using Cell-IQ Imagen 4.1.0software. Time-lapse images were stitched, analyzed to obtain wound area covered (%), length covered (μm) calculated with Cell-IQ Analyser 4.4.0 software, and used to plot a line graph as means of percent area covered by U2OS wild-type and Caldesmon KO cells from three individual repeats.

## 3D-matrigel invasion assay

The 3D-matrigel assay was used to assess the invasive property of wild-type and Caldesmon KO U2OS cells as reported earlier[79]. Cell invasion was performed using 6.5 mm with 8.0 μm pore polycarbonate membrane insert (CLS3422-48EA, Sigma-Aldrich) as per manufacturer's guidelines. 500 μl of 300 μg/ml matrigel was added into the transwell insert without creating bubbles and allowed to polymerize for 30 min at 37 °C without letting it dry. 500 μl of serum-free DMEM media was added to the upper chamber of the insert while 20% FBS containing DMEM was added to the lower chamber, i.e., into the well such that the insert was immersed into the media. 25 ng/ml epidermal growth factor (AF-100-15, Peprotech) was added to the lower chamber as a chemoattractant. U2OS wild-type and Caldesmon KO cells were trypsinized, suspended in 1.5% FBS containing DMEM media, added $5 \times 10^4$ cells to the upper chamber, and incubated for 6 h at 37 °C. Following incubation, the inserts were removed from wells, washed with PBS, scrapped with moist cotton swabs from inside the insert to remove cells, fixed (4% paraformaldehyde), permeabilized (0.1% Triton X-100), and counterstained with DAPI and phalloidin 488 dyes. The insert was cut, mounted on a slide on Vectashield® Vibrance™ antifade mounting medium (H-1700-10, Vector Laboratories), and images were captured with Leica DM5000B upright microscope equipped with ×20/0.7 HC PL APO CS wd = 0.59 objective, DAPI-5060C (ex 377/50, em 447/60) and GFP-4050B (ex 466/40, em 525/50) filters and Hamamatsu Orca-Flash4.0 V2 sCMOS camera. Minimum five images across different view-fields captured for wild-type and Caldesmon KO cells from three replicates. The images were visualized using LAS X 3.4.2 and processed using Fiji imageJ 1.53c software.

## Kymograph analysis

To analyze the fluctuations in myosin filaments in wild-type and Caldesmon-depleted U2OS cells, myosin filaments in transverse arcs flowing from the cell periphery towards the cell center were tracked for 30 min. Time-lapse images of selected transverse arcs collected at 100 s intervals were applied to draw the kymographs.

## DiAna co-localization analyses

To determine the percent co-localization volume and distance analysis between two protein molecules in a three-dimensional space, Fiji imageJ 1.53c-based DiAna was utilized as described previously[80]. 3D-imageJ suite is essential for executing DiAna tool. For DiAna colocalization analysis, GFP- and mCherry-tagged protein images were captured using GE Deltavision OMX super-resolution microscope as described in the super-resolution imaging section. SIM images constructed from average intensity projections of Z-stacks were used as source images in DiAna plugin. Green and red channel images were split, and segmented with the 3D-iterative segmentation tool in imageJ using the following criteria; min vol. pixels: 10, max vol. pixels: 100, min threshold/contrast: 10, elongation criteria (for F-actin), step threshold method with best segmentation and method value: 100. The segmented images were further uploaded in DiAna "analyze" option. % volume colocalization and distance between one or more red and green dots were measured from center-to-center, center-to-edge, and edge-to-center of red and green dots. The mean volume co-localized (%) and distance (μm) between two protein molecules in different groups were calculated and plotted. The complete tutorial is available online: [https://docs.google.com/presentation/d/e/2PACX-1vTcw7Zsc8gX4TsU1pwMsRlSVwVOa89sUQaGEwxWeANZsA3DCEwCU2PSvKruTVoN6V5D3DaDnNHI4-Yx/pub?start=false&loop=false&delayms=3000&slide=id.p].

## Statistics and reproducibility

Single-blind analysis of different groups was performed for the following experiments: Traction force microscopy (Fig. 4a, b and Supplementary Fig. 5d, e); live-cell imaging of retrograde arc flow (Fig. 4c, d and Supplementary Fig. 5f, g); laser ablation studies (Fig. 4h–k and Supplementary Fig. 6a–g). Double-blind imaging and analysis were performed for the following experiments: NM-IIA filament phenotype in wild-type and Caldesmon KO U2OS cells (Fig. 5a–c); NM-IIA filament rescue phenotype in Caldesmon FL-and domain-null expressing cells (Fig. 6a, b). The YAP localization was quantified as reported previously[44,81]. For analyzing the YAP localization in U2OS wild-type and Caldesmon KO cells (Fig. 4e–g), YAP expression in cell-cytoplasm and nucleus was quantified using Fiji ImageJ-ROI tool blindly for wild-type and Caldesmon KO cells. Thereafter, the ratiometric expression of YAP in the nucleus: cytoplasm was calculated by diving YAP-nuclear expression to YAP-cytoplasmic expression. Other experiments followed non-random execution. Densitometric quantifications of Western blots and co-sedimentation assay gels were carried through Image Lab™ 6.0 (Biorad) and Fiji imageJ 1.53c software. In line plot analysis, the highest mean fluorescence intensity value was set to 1 AU within the group, and the rest intensity values were normalized to that value. In Western blots, the mean values after quantifications from three experiments were normalized with wild-type protein levels, the mean value of wild-type data was set to one, and the intensities of other groups were normalized to that value. Experiments described in Figs. 1a–d, 3a, g, 6c, 7a, 8a, and Supplementary Figs. 1a–c, 2a, c–d, 3a–e, 5b, f, g, 7b, i, 8a–c, 9a, 10, 11a–c were repeated at least in three independent experiments, and we observed similar results as shown in the figures in every individual repeat. All the values obtained after quantifications were recorded, analyzed, and calculated in spreadsheets using Microsoft excel version-2016. The statistical analysis was performed using GraphPad Prism 7.03 software. Statistical significance between two groups was assessed by unpaired, non-parametric one-tailed/ two-tailed $t$-tests or by one-way ANOVA while, two-way ANOVA was implemented to compare more than two independent groups. The $p$-values were adjusted for multiple comparisons and indicate 0.05 significance with 95% CI. Significance was determined with Tukey's post hoc HSD analysis and Mann–Whitney/Wilcoxon rank-sum test (MWW). The graphs indicate the $p$-values for respective analyses.

## Reporting summary

Further information on research design is available in the Nature Research Reporting Summary linked to this article.

## Data availability

The data supporting the findings of the study are available in the manuscript and supplementary information. Other raw data generated in the study are available from the corresponding author on reasonable request. The statistical data generated in this study are provided in the Source Data files. Source data are provided with this paper.

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

## Acknowledgements

This work is supported by grants from the Sigrid Jusélius Foundation (4708344) and Academy of Finland (302161 and 346133) (to P.L.), NIH R01GM122375 grant (to S.K.), and NSF Graduate Research Fellowship Program (to V.D.T.). We thank the technical staff at the Institute of Biotechnology Light Microscopy Unit (LMU) and Biomedicum Imaging Unit (BIU) for the technical support in microscopy and image analysis. We also thank Mirva Tirkkonen for technical assistance. Confocal and 2-photon imaging experiments for the laser nanosurgery were conducted at the CRL Molecular Imaging Center at UC Berkeley, RRID:SCR_017852, supported by NSF DBI-1041078. We thank Holly Aaron and Feather Ives for their microscopy advice and support.

## Author contributions

P.L. and K.C. crafted the original idea and initiated the study. S.B.K. designed and performed the majority of the experiments, and analyzed and interpreted the results. S.K. and V.D.T. designed and performed the

laser ablation experiments, and K.K. contributed to cloning and protein purification. S.T. performed the analysis of TFM data, and R.K. contributed to the image analysis. U.E. carried out initial super-resolution microscopy experiments. S.B.K. and P.L. wrote the manuscript with input from all the authors.

## Competing interests

The authors declare no competing interests.
