## [Peer Review File · Nature Communications]

REVIEWER COMMENTS

Reviewer #1 (Remarks to the Author):

The authors present a detailed study of the spatial distribution and putative role of the actin/myosin-binding molecule caldesmon for stress fibers in non-muscle cells. While the spatial organization and molecular components of striated muscle are well known, this is much more unclear for these contractile actomyosin bundles formed by animal cells, despite the fact that they are essential for cell mechanics, cell shape, cell migration and mechanosensing. Earlier different hypotheses have been suggested regarding the putative role of caldesmon in non-muscle cells, but the authors show that the traditional view (restriction of force generation) is not correct. Rather the main function seems to be connecting the myosin neck to a subset of the actin filaments in the bundle (namely those not singled out by tropomyosin as tracks for myosin), thereby suppressing lateral sliding of the myosin minifilaments along the stress fibers. This leads to a disordered myosin array and less force in caldesmon KOs.

Overall, this study is very solid, convincing and interesting. The authors have uncovered a new and formerly unknown aspect of stress fibers and this is an important advance towards a complete understanding of their function. I note however that the described results are very focused on one molecule and one might wonder how to achieve a more complete picture. In particular, I note that Ref. 51 by Hu and Bershadsky in *Nature Cell Biology* 2017 (Long-range self-organization of cytoskeletal myosin II filament stacks) uncovered other aspects of stress fibers organization that are not discussed here. That work had a strong focus on α -actinin and the capping proteins, but otherwise was very similar in the approach (mainly SIM of appropriately marked proteins). In my view, the current work could become more complete by also looking into α -actinin and capping proteins. As far as I can see, α -actinin is mentioned only in passing in the supplement and it would be nice to show some line plots how it co-localizes with caldesmon. My expectation would be that α -actinin should show a clear complementary localization to caldesmon, and it would be interesting to see how its spatial distribution is perturbed in the caldesmon KO. Such a result should go into the main text. Likewise one should also clarify if the capping proteins are being affected. Then the model in Fig. 6 would become more complete.

The authors show a very specific location for caldesmon, which is a huge protein with many other binding sites. In addition, they show that it is stretched, which opens up the perspective of cryptic binding sites, as known from talin and vinculin. Do the authors think that this implies some kind of mechanosensitivity here? Could the authors say more on the downstream effects of caldesmon binding exactly in this location and in a stretched state?

The authors conclude that caldesmon prevents lateral sliding of myosin, but in my view this conclusion is very indirect. Could this be shown more directly with kymographs? It has been shown by the group of Tanmay Lele that stress fibers show fluctuations of their minifilaments and I wonder if the authors could confirm this and show how it is changed in the caldesmon KO?

Reviewer #2 (Remarks to the Author):

The manuscript by Kokate et al examines the role of caldesmon in cells. From biochemical experiments performed mostly with smooth muscle proteins, this protein was proposed to be a calcium-dependent regulator of actin-myosin interaction, functioning somewhat analogously to the troponin-TM system in skeletal muscle. The current study uses super resolution light microscopy, cell migration assays, traction force microscopy and laser

ablation of stress fibers to quantify actomyosin function in wild type and knock-out cells. They come to the conclusion that caldesmon does not function as a negative regulator of actomyosin function in cells, but instead forms a structural role to insure the alignment of myosin filaments with the actomyosin bundles. The study is well constructed and controlled. I could not open the first few movies that were not in an .avi format. I have a few questions.

I don't understand the caldesmon binding experiments. The Y-axis suggest that 100 μM of caldesmon is bound, and yet only 1 μM is present in the assay. Please explain.

Biochemical experiments demonstrated that calcium + calmodulin relieved the caldesmon-induced inhibition of actomyosin ATPase. What role do you envision for calcium/calmodulin in your models?

It would be interesting to see whether the full length h-caldesmon could rescue the KO phenotypes. As the authors mention, the difference between these two paralogs is the length of central spacer region. Could this be optimized for the different type of myosin filaments (side polar vs short bipolar) seen in smooth muscle vs nonmuscle cells?

How is the caldesmon localization affected by depolymerization of actin?

POINT-BY-POINT RESPONSES TO REVIEWERS' COMMENTS

Reviewer #1

The authors present a detailed study of the spatial distribution and putative role of the actin/myosin-binding molecule caldesmon for stress fibers in non-muscle cells. While the spatial organization and molecular components of striated muscle are well known, this is much more unclear for these contractile actomyosin bundles formed by animal cells, despite the fact that they are essential for cell mechanics, cell shape, cell migration and mechanosensing. Earlier different hypotheses have been suggested regarding the putative role of caldesmon in non-muscle cells, but the authors show that the traditional view (restriction of force generation) is not correct. Rather the main function seems to be connecting the myosin neck to a subset of the actin filaments in the bundle (namely those not singled out by tropomyosin as tracks for myosin), thereby suppressing lateral sliding of the myosin minifilaments along the stress fibers. This leads to a disordered myosin array and less force in caldesmon KOs. Overall, this study is very solid, convincing and interesting. The authors have uncovered a new and formerly unknown aspect of stress fibers and this is an important advance towards a complete understanding of their function.

We thank the reviewer for positive feedback and for excellent suggestions to improve our study.

I note however that the described results are very focused on one molecule and one might wonder how to achieve a more complete picture. In particular, I note that Ref. 51 by Hu and Bershadsky in Nature Cell Biology 2017 (Long-range self-organization of cytoskeletal myosin II filament stacks) uncovered other aspects of stress fibers organization that are not discussed here. That work had a strong focus on α -actinin and the capping proteins, but otherwise was very similar in the approach (mainly SIM of appropriately marked proteins). In my view, the current work could become more complete by also looking into α -actinin and capping proteins. As far as I can see, α -actinin is mentioned only in passing in the supplement and it would be nice to show some line plots how it co-localizes with caldesmon. My expectation would be that α -actinin should show a clear complementary localization to caldesmon, and it would be interesting to see how its spatial distribution is perturbed in the caldesmon KO. Such a result should go into the main text.

Response: *We appreciate reviewer 1 for these valuable suggestions. We have now addressed this question by performing additional 3D-SIM experiments on wild-type and caldesmon knockout cells. These experiments revealed that α -actinin displays a periodic pattern in U2OS cells. Caldesmon doublets (visualized by using mCherry-caldesmon) localize in between α -actinin peaks in wild-type cells, as expected from the localization pattern of caldesmon in respect to myosin II. On the other hand, the α -actinin distribution in caldesmon knockout cells was somewhat less regular. Whereas the distance between α -actinin peaks in wild-type cells was $\sim 0.6 \mu\text{m}$, the distances between α -actinin peaks in the caldesmon knockout cells were typically shorter and more variable. This indicates that also the 'sarcomeric' organization of actin filaments within stress fibers is somewhat disturbed in the caldesmon knockout cells. This is most likely a result of aberrant lateral movement of myosin II bundles, because there is no evidence of caldesmon interacting with α -actinin. The new results are shown in Fig. 7A-E, and discussed in the manuscript text on page 9 (lines 248-254) and on page 12 (lines 355-361).*

We also examined the distributions of heterodimeric capping protein and tropomodulin-1 in these cells. Our results show that capping protein (eGFP-CP-62) does not localize to stress fibers in U2OS cells, but instead accumulates to the lamellipodium and sites of endocytosis as previously reported (e.g. Hakala et al., 2021). These new data are shown in Fig. S10 and discussed on page 9, lines 256-258. Tropomodulin-1, on the other hand localized to stress fibers, although its distribution within these structures in U2OS cells was not as regular as previously shown for tropomodulin-3 in REF52 cells by Hu et al., (2017). Please note that we used here a GFP-tropomodulin-1 construct, because tropomodulin-1 specifically localizes to the contractile stress fibers in U2OS cells (Kumari et al., 2020). Our 3D-SIM experiments demonstrated that tropomodulin-1 often localized close to the center of caldesmon doublets, as expected from the distribution of caldesmon in respect of myosin II. Similarly to the α -actinin localization, the tropomodulin localization in the caldesmon knockout cells was

less regular and the distances between the tropomodulin peaks along the stress fibers were more variable and shorter, again indicating that the 'sarcomeric' organization of actin filaments of stress fibers was also somewhat disrupted in the absence of caldesmon. These new data are shown in Fig. S9, and discussed in the text on page 9 lines 254-259 We have also edited the 'Discussion' (page 12, lines 355-361) and the model figure (Fig. 8D) to better illustrate that also the actin filament organization is somewhat disrupted in caldesmon knockout cells.

The authors show a very specific location for caldesmon, which is a huge protein with many other binding sites. In addition, they show that it is stretched, which opens up the perspective of cryptic binding sites, as known from talin and vinculin. Do the authors think that this implies some kind of mechanosensitivity here? Could the authors say more on the downstream effects of caldesmon binding exactly in this location and in a stretched state?

Response: *Caldesmon is a large protein and likely to interact also with many other proteins in addition to tropomyosin-actin and myosin II. An earlier in vitro study by Furst et al., (1986) also demonstrated that caldesmon is an elongated and relatively flexible protein, and thus it is indeed possible that it also functions as a some kind of mechanosensor in stress fibers. On the other hand, our FRAP experiments provided evidence that caldesmon displays very rapid dynamics in stress fibers, and its turnover was not affected by myosin inhibition (Fig. 1B-F). Therefore, if caldesmon also functions as a mechanosensor, it is most likely able to respond to forces only in a very short time-scale. This is now discussed on page 13, lines 365-370.*

The authors conclude that caldesmon prevents lateral sliding of myosin, but in my view this conclusion is very indirect. Could this be shown more directly with kymographs? It has been shown by the group of Tanmay Lele that stress fibers show fluctuations of their minifilaments and I wonder if the authors could confirm this and show how it is changed in the caldesmon KO?

Response: *To address this concern, we conducted kymograph analysis from movies of wild-type and caldesmon knockout cells. This analysis illustrates that myosin filaments displayed a relatively uniform retrograde flow with only small lateral fluctuations in wild-type cells, whereas in caldesmon knockout cells myosin filaments displayed abnormal lateral fluctuations, resulting in the formation of dense myosin 'clumps'. These new analyses are shown in Fig. S11A-B.*

Reviewer #2:

The manuscript by Kokate et al examines the role of caldesmon in cells. From biochemical experiments performed mostly with smooth muscle proteins, this protein was proposed to be a calcium-dependent regulator of actin-myosin interaction, functioning somewhat analogously to the troponin-TM system in skeletal muscle. The current study uses super resolution light microscopy, cell migration assays, traction force microscopy and laser ablation of stress fibers to quantify actomyosin function in wild type and knock-out cells. They come to the conclusion that caldesmon does not function as a negative regulator of actomyosin function in cells, but instead forms a structural role to insure the alignment of myosin filaments with the actomyosin bundles. The study is well constructed and controlled. I could not open the first few movies that were not in an .avi format. I have a few questions.

We thank the reviewer for valuable comments and suggestions to improve our manuscript. We also apologize that the movies were originally not in a uniform format, and have now made all movies in the avi-format.

I don't understand the caldesmon binding experiments. The Y-axis suggest that 100 uM of caldesmon is bound, and yet only 1 uM is present in the assay. Please explain.

Response: *We thank the reviewer for pointing out this mistake. The Y-axis shows the percentage of Caldesmon bound to F-actin. We have now corrected this error in all relevant panels.*

Biochemical experiments demonstrated that calcium + calmodulin relieved the caldesmon-induced inhibition of actomyosin ATPase. What role do you envision for calcium/calmodulin in your models?

Response: *This is an interesting question. Earlier studies provided evidence that the binding sites of actin and calmodulin overlap on caldesmon, and thus it is possible that calcium/calmodulin would release caldesmon from stress fibers (this is because our experiments in Fig. S2 provided evidence that actin-binding is critical for strong stress fiber association of caldesmon). To study the possible role of calcium/caldesmon in regulating caldesmon localization, dynamics or function in stress fibers, one would need to identify specific mutants that inhibit the interaction of caldesmon with calmodulin without affecting its actin-binding. This is now discussed on page 13, lines 362-365.*

It would be interesting to see whether the full length h-caldesmon could rescue the KO phenotypes. As the authors mention, the difference between these two paralogs is the length of central spacer region. Could this be optimized for the different type of myosin filaments (side polar vs short bipolar) seen in smooth muscle vs nonmuscle cells?

Response: *This is an interesting point, and we have now performed additional rescue experiments using H-Cald construct. These experiments provided evidence that H-Cald rescues the knockout phenotype quite well, but not to the same extent as the L-Cald. Because the high molecular weight isoform (H-Cald) expressed in smooth muscle cell is identical to L-Cald (expressed in non-muscle cells), except for harboring a longer central repeating sequence of charged amino acids, these data indicate that length of caldesmon molecule may be fine-tuned to match the myosin II isoform expressed in the corresponding cell-type. These new data are shown in Fig. S11C-D, and discussed in the manuscript text on page 10, lines 284-289.*

How is the caldesmon localization affected by depolymerization of actin?

Response: *To explore the effects of actin depolymerization on the localization of caldesmon, we first treated wild-type U2OS cells with actin polymerization inhibitors Latrunculin A (LatA) and Cytochalasin D (1 μ M, 60 min). Immunofluorescence microscopy revealed that with these high actin inhibitor concentrations all actin filaments structures, including stress fibers, were almost completely disturbed, and consequently caldesmon no longer displayed any specific localization in cells (data not shown). We did not include these data to the revised manuscript, because they just confirm that caldesmon binds actin and that actin-actin-binding is important for cellular localization of this protein (this was already demonstrated by the experiments presented in Fig. S2). Therefore, we also treated wild-type U2OS cells with 0.2 μ M LatA for 15 min to partially depolymerize dynamic actin filament structures. These experiments, shown in Fig. S1C and discussed on page 4, lines 102-104, demonstrate that caldesmon still localizes to stress fibers after a 'more gentle' LatA-treatment, providing evidence that diminishing actin filament assembly does not disrupt localization of caldesmon to stress fibers.*

REVIEWERS' COMMENTS

Reviewer #1 (Remarks to the Author):

The authors have answered very well to my comments (and in my view also to the ones of the other reviewer). In particular, new Fig. 7 on the role of α -actinin is a very important new element which allows the reader to get a better systems understanding. Overall this study is highly interesting and an important advance in our understanding of how stress fibers in non-muscle cells regulate force generation. I note that I had problems with the movies, especially the ones on the wound healing assays, they should be checked again.

Reviewer #2 (Remarks to the Author):

I am satisfied with the authors' revision in response to my comments. This is an interesting study that will make an important contribution to the field